# Assisting students' writing with computer-based concept map feedback: A validation study of the CohViz feedback system

Christian Burkhart[1]*, Andreas Lachner[2], Matthias Nückles[1]

1 Department of Educational Science, University of Freiburg, Freiburg, Germany, 2 Department of Education, University of Tübingen, Tübingen, Germany

* christian.burkhart@ezw.uni-freiburg.de

**Data Availability Statement:** The CohViz system is available on the Open Science Framework (OSF) under the following address: https://doi.org/10.17605/OSF.IO/SA53E. All data files and analyses of

## Abstract

CohViz is a feedback system that provides students with concept maps as feedback on the cohesion of their writing. Although previous studies demonstrated the effectiveness of Coh-Viz, the accuracy of CohViz remains unclear. Thus, we conducted two comprehensive validation studies to assess the accuracy of CohViz in terms of its reliability and validity. In a reliability study, we compared the concept maps generated by CohViz with concept maps generated by four human expert raters based on a text corpus comprising students' explanatory texts ($N = 100$). Regarding the depiction of cohesion gaps, we obtained high accordance between the CohViz concept maps and the concept maps generated by the human expert raters. However, CohViz tended to overestimate the number of relations within the concept maps. In a validity study, we examined the validity of CohViz and compared central features of the CohViz concept maps with convergent linguistic features and divergent linguistic features based on a Wikipedia text corpus ($N = 1020$). We found medium to high agreement with the convergent cohesion features and low agreement with the divergent features. Together, these findings suggest that CohViz can be regarded as an accurate feedback system to provide feedback on the cohesion of students' writing.

## Introduction

Comprehensible writing can be regarded as a crucial writing skill in today's knowledge society [1–3]. A central feature that contributes to the comprehensibility of a text is cohesion [4,5]. Cohesion refers to grammatical or lexical devices which explicitly relate ideas within and across text segments [6]. It can be established in two ways. On a local level, writers can improve the cohesion of their texts by relating information between neighboring sentences, for instance by reiterating arguments or using near-synonyms (i.e., argument overlap) [6,7]. On a global level, students can improve the cohesion of their texts by explicitly relating text passages [8], for instance, by adding headings or by following a particular rhetorical structure [1,9]. Despite its central role in supporting readers' comprehension, college students often face difficulties in writing cohesive texts [10,11]. Therefore, particularly students require ample formative

this manuscript are available on the OSF under the following address: https://doi.org/10.17605/OSF.IO/UHPF3.

**Funding:** The authors received no specific funding for this work. The article processing charge was funded by the Baden-Wuerttemberg Ministry of Science, Research and Art and the University of Freiburg in the funding programme Open Access Publishing.

**Competing interests:** The authors have declared that no competing interests exist.

feedback on the cohesion of their writing [12]. Providing instant feedback, however, is relatively time-consuming and often not feasible during regular teaching. Thus, a variety of computer-based feedback systems have recently been developed to improve students' writing for specific linguistic features such as text cohesion, particularly in the early stages of writing instruction [12–15]. The advantage of these systems is that they provide students with instant and time-independent information about the quality of their writing. Many of these systems generate graphical visualizations from students' texts in the form of concept maps [16,17]. These concept maps provide students with an additional external representation of their text and direct their attention to distinct textual deficits to activate appropriate revision activities [15]. In the current paper, we provide a thorough validation study of a promising graphical feedback system, CohViz, which was explicitly designed to improve the cohesion of students' writing. In two studies we investigated whether CohViz accurately provides feedback. Therefore, we measured the reliability and the validity of the generated feedback information using two corpus-based studies. Such a validation study about the accuracy of feedback of CohViz is particularly important, as recent studies provided evidence that the effectiveness of feedback largely depends on the accuracy of the provided information within the feedback [18,19].

## CohViz: A feedback system to support students' cohesive writing

CohViz is a graphical feedback system we developed which is freely available online [20]. CohViz automatically provides students with concept maps as external representations of their texts to improve the cohesion of their writing (see Fig 1) [15,21,22]. As such, using CohViz has been demonstrated to be feasible in online or blended learning settings [15,23] but also in large lecture classes [22] with high levels of students' self-regulation during writing. By combining distinct state-of-the-art natural language processing (NLP) technologies [24–26], CohViz automatically generates concept maps from students' self-written texts (see Fig 1). Local cohesion deficits are highlighted by colored fragments, i.e. sub-concept maps that are not related to other concept maps. Global cohesion deficits are visualized in terms of the text's macrostructure [27], i.e. by the way the concepts of the texts are used and related. The key advantage of CohViz is that it supplements teachers' feedback in that it allows to provide students with quick and specific feedback on the degree of cohesion of their texts. In practice, the feedback can be embedded as a homework assignment [22] or modeled by teachers to show textual deficits of non-cohesive texts [15]. For example, an instructor could use the tool to inform students why a particular text is high or low in cohesion (e.g., by showing texts which yield numerous fragments or texts which do not have central concepts with many relations). Similarly, students can use the tool to check short texts such as abstracts or summaries for cohesion during writing.

## Generation of the CohViz concept maps

To generate the concept maps, CohViz uses a four-step procedure (see Fig 2). First, CohViz segments each text into sentences (i.e., the segmentation phase). Second, in the extraction phase, CohViz determines the grammatical function (i.e., subject, direct object, indirect object) of each concept using the RFTagger [25], and reduces the identified concepts to lexical lemmas (e.g., *theory* instead of *theories*) to improve comparisons across concepts [24]. Third, in the relation phase, CohViz extracts relationships between concepts according to their grammatical relationship within sentences [28], and co-references between sentences: Within sentences, a relation depicts the semantic relationship between two concepts in terms of distinct propositions. Each proposition encompasses two arguments which are grammatically related (i.e., subject, object, direct object). To reduce the complexity of the concept maps, the type of

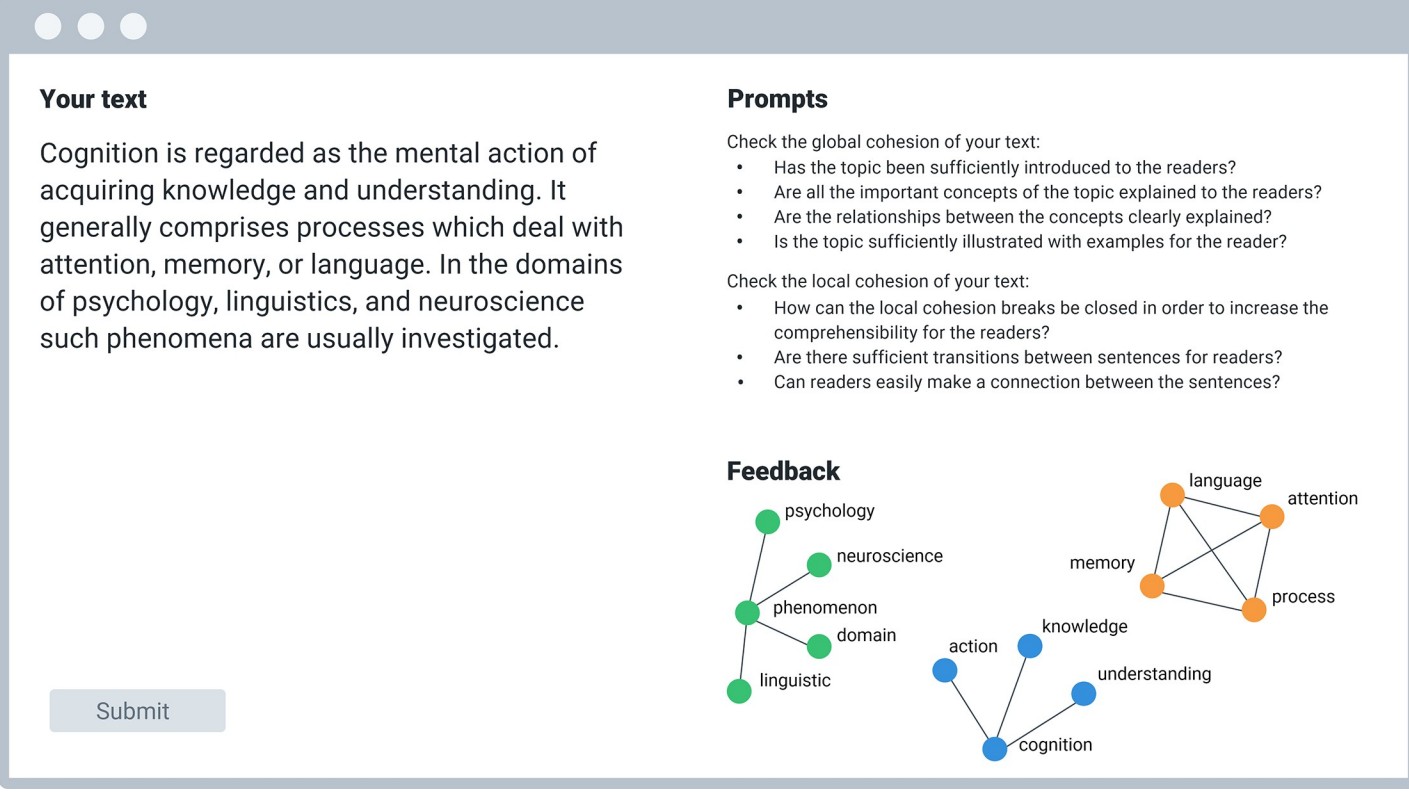

**Fig 1. Example of a concept map generated from CohViz.** Colored fragments indicate concepts that are not related to the rest of the concepts and are thus indicative of a problem in cohesion. For example, the writer did not relate the first two sentences, leading to a fragmented concept map. The concept map contains three fragments, 13 concepts, and 13 relations.

relation (i.e., its predicate) is not visualized. For example, for the sentence "Hollywood baked the rolls", CohViz builds one relation. The subject "Hollywood" is combined with the object "roll". Hence, CohViz represents semantic information between arguments in terms of partial propositions in which the predicate is not explicitly labeled (see also, [17,29] for related approaches). Additionally, CohViz visualizes unambiguous (but not ambiguous) co-references. Unambiguous pronouns refer to co-references which can be matched to concepts of the previous sentences (e.g., unambiguous co-reference: "**The sailors** boarded the ship. **They** were ready to start."; ambiguous co-reference: **The sailors** boarded the ship of the companions. **They** were ready to start). The relations are then stored as word-pairs and are visualized in the form of a concept map.

## Previous research on the effectiveness of CohViz

So far, the effectiveness of CohViz has been tested in various settings between 2017 and 2019, including controlled laboratory studies and ecologically valid field studies in a wide range of disciplines (e.g. biology, ethics, philosophy, educational psychology, teacher education). In these experimental studies, the authors examined the effectiveness of CohViz with college students [15,21,22,30]. Overall, in a mini meta-analysis Burkhart, Lachner, and Nückles [31] could show that CohViz has a medium effect on both local and global cohesion and was thus effective in improving the cohesion of students' texts (i.e., local cohesion $g = 0.62$; global cohesion $g = 0.57$). A potential explanation for the obtained effects of CohViz can be found in the think-aloud study by Lachner, Burkhart, and Nückles [15]. The think-aloud study could show

**0** ORIGINAL

Cognition is regarded as the mental action of acquiring knowledge and understanding. It generally comprises processes which deal with attention, memory, or language. In the domains of psychology, linguistics, and neuroscience such phenomena are usually investigated.

**1** SEGMENTATION

1. Cognition is regarded as the mental action of acquiring knowledge and understanding.
2. It generally comprises processes which deal with attention, memory, or language.
3. In the domains of psychology, linguistics, and neuroscience such phenomena are usually investigated.

**2** EXTRACTION

1. cognition [subject]; action [object]; knowledge [object]; understanding [object]
2. process [object]; attention [object]; memory [object]; language [object]
3. phenomena [subject]; domain [object]; psychology [object]; neuroscience [object]; linguistic [object]

**3** RELATION

cognition  - understanding memory - language
cognition  - knowledge memory - attention
cognition  - action phenomena - domain
process    - language phenomena - psychology
process    - memory phenomena - neuroscience
process    - attention phenomena - linguistic
attention  - language

**4** MERGING

**Fig 2. Depiction of the four step procedure of the CohViz feedback.**

that students who processed the concept maps could infer both local and global writing plans from the concept maps directly. Besides, the analysis of the concept map triggered negative monitoring processes (i.e., student thought about the macrostructure of a text) which led to further planning processes.

These findings may be indicative of the prognostic validity of CohViz. However, it is less clear whether the information provided by the feedback is accurate. Research on the accuracy is important, as research indicates that the effectiveness of feedback depends on its accuracy [18,19]. Therefore, in-depth research on the accuracy is mandatory for potential future advancements of such feedback systems.

### Overview of the current study

Against this background, we conducted two corpus-based studies in which we investigated the accuracy of the CohViz feedback in terms of its reliability and validity (see Fig 4).

In the reliability study, we investigated the reliability of CohViz concept maps compared to concept maps based on propositional segmentations by human expert raters on a corpus of 100 expository texts of novice writers collected from previous experimental studies. To compare the concept maps, we used structural indicators commonly used in research on concept maps [32–35]. We assumed that CohViz concept maps yield similar results in the number of concepts, relations, and fragments compared to the human expert raters.

Regarding the validity of CohViz, we assessed the convergent and divergent validity of the CohViz system with automated measures from well-established assessment systems [36–39], as used for example by the well-known system Coh-Metrix [40]. As CohViz was predominantly designed to provide feedback on the cohesion of students' writing, we focused on the number of fragments as the central indicator of cohesion provided by CohViz and compared it with convergent (i.e., argument overlap, semantic overlap) and divergent (i.e., syntactic complexity, lexical diversity, and word concreteness) linguistic features of text cohesion. To benchmark the fragments against the convergent and divergent measures, we used a representative text corpus comprising 1020 expository texts from the German Wikipedia. Regarding the convergent validity, we expected medium to large correlations between the number of CohViz fragments as an indicator of text cohesion and the convergent features of text cohesion (i.e., argument overlap, semantic similarity). Regarding the divergent validity, we assumed no substantial correlations between the CohViz fragments and features on the syntactic and lexical level.

## Methods

### Corpora collection

**Reliability corpus.** The corpus to test the reliability of CohViz was compiled by the authors as a sample from an entire set of 901 expository texts written in German by college students. All texts were produced by novice writers in the course of different experimental studies conducted by the authors and were conducted between 2015 and 2018 at German universities. We made sure to compile texts with a representative range of topics in the natural sciences and the humanities. In all of these studies the dependent variables of interest were measures of text quality (e.g., local and global cohesion). In those studies, students had 15 minutes to accomplish a writing task. For each of these studies, students gave their written consent for the scientific use of the data. As manual segmentation and coding of the entire corpus of 901 texts would be enormously time-intensive, we decided to use a random, yet a representative subset of 100 texts from the entire text corpus (see Table 1). To test whether the sample was large enough to find small to medium effects, we conducted a power analysis with a sample size of

**Table 1. Descriptive statistics of the corpus of the reliability study.**

| Topic | | N | Mean number of words | Flesch-Kincaid readability score | Mean number of sentences |
|---|---|---|---|---|---|
| All texts | | 100 | 200.47 (72.14)[a] | 12.5 (2.09) | 12.42 (5.02) |
| Natural sciences | | | | | |
| | Combustion engines | 10 | 261.3 (66.32) | 10.43 (1.35) | 19.2 (5.61) |
| | Osmosis | 10 | 136.9 (18.84) | 11.03 (1.39) | 9.0 (2.83) |
| | Natural selection | 10 | 166.3 (28.04) | 11.429 (1.77) | 10.0 (2.36) |
| | Data preservation | 10 | 234.6 (73.66) | 13.64 (1.73) | 12.5 (3.98) |
| Humanities | | | | | |
| | School systems | 10 | 212.9 (82.99) | 15.16 (2.13) | 11.5 (3.75) |
| | Knowledge representations | 10 | 204.5 (51.11) | 12.37 (2.26) | 14.0 (4.22) |
| | Cognitive Load Theory | 10 | 255.8 (90.78) | 12.63 (1.09) | 15.9 (6.01) |
| | Reading skills | 10 | 124.2 (19.25) | 13.37 (1.80) | 7.1 (1.10) |
| | Instructional explanations | 10 | 193.6 (43.66) | 12.79 (2.08) | 11.7 (2.63) |
| | Formative assessment | 10 | 214.6 (44.61) | 12.01 (1.07) | 13.3 (4.39) |

[a]Values in brackets refer to standard deviations.

100, and effect size of $r > .3$ and Pearson correlation as the statistical test. We obtained a satisfactory power of .93, indicating that the sample size was large enough to detect small to medium effects for convergent validity. To reduce experimenter bias, the selection was carried out via a computer algorithm written by the authors that randomly selected ten texts per topic. For each topic within a domain the algorithm selected 10 expository texts resulting in a total set of 100 texts. We aimed at a text corpus with high linguistic variability, as we were interested in testing the reliability of CohViz under realistic conditions. That said, the text corpus varied on central linguistic dimensions per topic, such as the number of words, $F(9, 90) = 6.03$, $p < 0.01$, $\eta_p^2 = .38$ (large effect), the number of sentences, $F(9, 90) = 7.72$, $p < 0.01$, $\eta_p^2 = .43$ (large effect), and the Flesch-Kincaid readability scores, $F(9, 90) = 6.44$, $p < 0.01$, $\eta_p^2 = .39$ (large effect). The Flesch-Kincaid is a measure for calculating the potential comprehensibility of a text [41] and uses various linguistic features (i.e., number of total words, number of total sentences, number of total syllables) to which different weighting factors are assigned. Texts with a high Flesch-Kincaid score are difficult to read, whereas texts with a low readability score are easy to read. To increase the comprehensibility of the score, it is commonly standardized to grade levels. The obtained readability score of the reliability corpus thus suggests that the texts were suitable for 12[th]-grade students.

**Validity corpus.** As a crucial feature of validity is the degree of representativeness [42], we sought to construct a corpus representing a broad range of topics and writing styles in expository writing. One easily-accessible database of expository texts is Wikipedia. Currently, the German Wikipedia comprises 2,409,327 articles. Thus, to test the validity of CohViz, we extracted 1020 German Wikipedia entries from the overall Wikipedia corpus (see Table 2). As students' expository texts in common classroom studies were rather short, we only analyzed the extended summary sections of the articles. To extract texts from a wide variety of topics and writing styles and because of the enormous size of the Wikipedia corpus, we used a computer algorithm that randomly extracted the texts from the corpus.

## Construction of the concept maps

To compare the CohViz concept maps to concept maps generated from human expert raters, we asked four human expert raters to segment the texts from the reliability corpus into

**Table 2. Descriptive statistics of the corpus of the validity study.**

| Features | | *M* | *SD* |
|---|---|---|---|
| Number of words per sentence | | 16.23 | 3.66 |
| Number of syllables per word | | 1.96 | 0.18 |
| Number of sentences | | 12.85 | 3.59 |
| Number of words per text | | 208.36 | 71.80 |
| CohViz features | | | |
| | Number of fragments | 3.68 | 1.99 |
| | Number of concepts | 53.16 | 19.06 |
| | Number of relations | 158.50 | 233.43 |
| Convergent features on the cohesion level | | | |
| | Adjacent semantic overlap | .25 | .15 |
| | Text level semantic overlap | .21 | .10 |
| | Adjacent argument overlap | .27 | .17 |
| | Text level argument overlap | .23 | .13 |
| Divergent features on the syntactic level | | | |
| | Average length of longest dependency | 9.31 | 2.48 |
| | Average number of complex nominals per clause | 0.68 | 0.27 |
| Divergent features on the lexical level | | | |
| | Word concreteness | 1.07 | 0.35 |
| | Root type-token ratio | 9.36 | 1.32 |

propositions as a basis for the generation of the concept maps (see Fig 4 for the full processing of the corpus). Therefore, we looked for experienced raters with a solid background in linguistics who had previous experience in propositional segmentation. Among the raters who fulfilled these criteria, we asked four advanced master students with a major in applied linguistics or learning and instruction to analyze the corpus. All raters came from the same German university as the authors. Their mean age was 24 (*SD* = 2.94). They were already familiar with the procedure of propositional segmentation since it was part of their studies' curriculum. To ensure a uniform prior knowledge of the procedure, each rater was provided with multiple in-depth training sessions (five hours on average) on propositional segmentation and text cohesion. In these training sessions, raters were instructed on different cohesion strategies (e.g., argument overlap, connectives, bridging information). Additionally, they were trained in propositional segmentation with authentic practice material. Each rater independently segmented the texts from the corpus into propositions. To standardize the segmentation procedure, the raters were supported with a segmentation manual when segmenting the texts. First, they were instructed to split each text into sentences. Next, they were asked to divide each sentence into propositions consisting of a predicate and its arguments (e.g., subject, possessor, direct object) [11,43]. For instance, the sentence "Neil went to the moon" would be broken down into the following proposition: P(subject = Neil, object = moon, predicate = go) and transferred to the word-pair: Neil–moon. Additionally, the raters were asked to relate adjacent sentences when they were explicitly related by meaningful cohesive devices (i.e., argument overlap, co-references, superordinate concepts, subordinate concepts, bridging information, and connectives). Each proposition was entered into a spreadsheet with the id of the participant, the number of each sentence, and the subject and object of each proposition as two variables. From this list and to reduce errors by manual coding, we automatically calculated three structural indicators commonly studied in concept mapping research [34,35] using the CohViz algorithm (see Fig 1 for an example, see [44] for the full computer algorithm): Concepts

represented the concepts from the text, relations represented the relationships among these concepts based on the propositional segmentation and fragments represented clusters of concepts within a text which were not related to the rest of the text and should thus indicate potential breaks of text cohesion. To guarantee the consistency and representativeness of the human expert segmentations, we followed a two-step procedure. First, all four raters coded a subset of 30 expository texts from the corpus of 100 texts. Interrater reliability for this subset was good to excellent: number of fragments ($ICC$ = .76), number of relations ($ICC$ = .91), and number of concepts ($ICC$ = .97). Second, as the interrater agreement between the four raters was high, two of the four raters coded the rest of the entire text corpus. Interrater reliability between the two raters again was excellent for all indices: number of fragments ($ICC$ = .89), number of relations ($ICC$ = .99), and number of concepts ($ICC$ = .99). Differences between the two remaining raters were resolved by discussion.

To compare the structural indicators of the human expert raters to CohViz, we generated the concept maps from the same reliability corpus using the above mentioned CohViz algorithm. In contrast to the propositional segmentation by the human expert raters, the CohViz concept maps were generated by state-of-the-art natural language techniques [24–26]. From these concept maps we computed the same structural indicators, that is the number of concepts, the number of relations, and the number of fragments.

## Measures

**Structural indicators of the concept maps.** The generated concept maps yielded three structural indicators commonly used in concept mapping research [34,35]: The number of concepts in each text, the number of relations between these concepts, and the number of fragments (see Fig 1). Fragments are the main indicator of text cohesion in CohViz and depict distinctly colored sub-concept maps that are not related to the rest of the concept map. An increase in the number of fragments should indicate a decrease in local text cohesion.

Formally, the number of fragments ($F$) can be defined as the sum of all sub-concept maps in a cluster graph [45]: $F = \sum_{f=1}^{n} F_i(C, R)$. $C$ denotes all concepts within a sub-concept map, $C = \{c_1, c_2, c_3, \ldots, c_n\}$. For example, in Fig 3, the green fragment includes the concepts *psychology*, *neuroscience*, *phenomenon*, *domain*, and *linguistic*. $R$ denotes all relations within a sub-concept map, for example, $R = \{\{c_1, c_2\}, \{c_1, c_3\}, \ldots \{c_1, c_n\}\}$. In Fig 3, for example, the green fragment has four relations with five concepts. Fragments are disjoint from each other so that fragments do not share concepts. For example, the green and the orange fragment in Fig 3 do not share a common concept. Formally, this fact can be described by $C(F_1) \cap C(F_2) = \emptyset$. The full algorithm for calculating these structural indicators can be found in the public data repository of this study [44].

**Convergent and divergent measures of text cohesion.** To investigate the validity of CohViz, we benchmarked the three structural indicators obtained by CohViz to convergent and divergent measures of text cohesion commonly used in well-established assessment systems for writing quality [36,46]. Convergent validity can be assessed by comparing the fragments as the central measure of text cohesion in CohViz to other text cohesion measures (e.g., argument overlap, LSA-dependent cohesion measures) [39]. Divergent validity can be assessed by comparing CohViz to divergent measures of text cohesion (e.g., syntactic or lexical complexity as indicators of writing quality). Since a comprehensive coverage of all available linguistic features of texts would result in a confusing plethora of computer-based indicators (n > 200), we followed suggestions by McNamara, Crossley, and McCarthy [47] and MacArthur et al. [5] and selected convergent and divergent measures commonly associated with writing quality and applied these measures to the validity corpus.

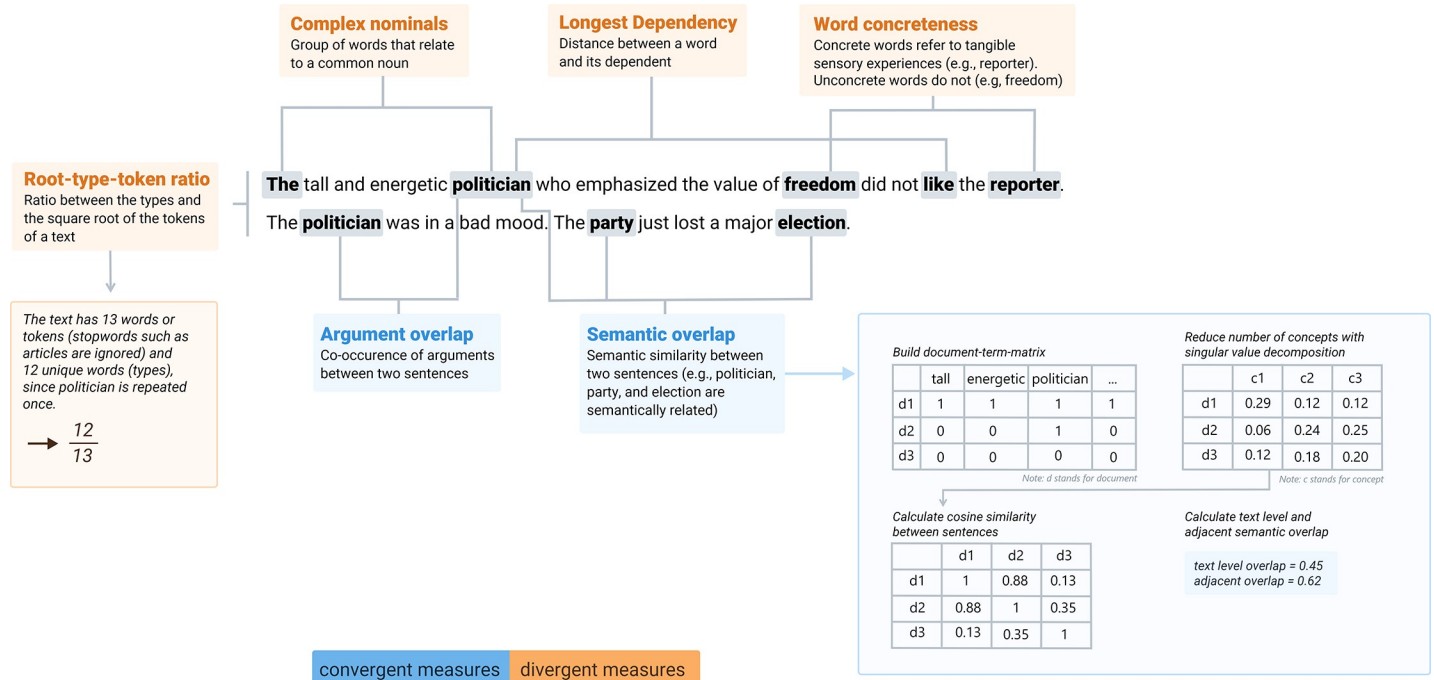

**Fig 3. Convergent and divergent features of the validity study.**

To obtain these convergent and divergent measures of text cohesion, we chose to analyze the texts with well-established assessment systems as they have been thoroughly studied in the past and represent the state-of-the-art tools in text-processing. As the Wikipedia entries were written in German we used the assessment tool developed by Hancke et al. [37], which has been designed for the German language. The system provides researchers with validated and commonly used linguistic indicators regarding the cohesion, syntactic, and lexical complexity of a text [4,5,48]. This system is comparable to Coh-Metrix, which was developed for the English language by Graesser et al. [46]. To compare these measures with CohViz, we first calculated the structural indicators from the validity corpus and then extracted the convergent and divergent validity measures from the validity corpus with the assessment system developed by Hancke et al. [37].

*Convergent features on the cohesion level.* Argument overlap is a central indicator of cohesion and measures the co-occurrence of arguments between two sentences [6]. To measure argument overlap, texts are first segmented into sentence pairs. Afterward the sentence pairs, which contain a common argument (e.g., noun phrase), are counted and divided by the total amount of sentence pairs (see Fig 3 for an example). There are two versions of argument overlap.

Adjacent argument overlap measures the overlap of arguments between neighboring sentences. Formally, adjacent argument overlap is defined as the ratio between the sum of the adjacent sentence pairs that share a common argument $P_{i,i+1}$, divided by all adjacent sentence pairs in a text $(n-1)$:

$$Adjacent\ argument\ overlap = \sum_{i=1}^{n-1} P_{i,i+1} * \frac{1}{n-1}$$

Text level argument overlap measures the overlap of each sentence with every other sentence in a text and is a proxy for the global cohesion of a text. Formally, text level overlap is defined as the ratio of the sum of all sentence pairs within a text that share a common argument $P$, divided by all possible sentence pairs within a text $\left(n * \frac{n-1}{2}\right)$. To calculate text level argument overlap, the sentences are stored in a lower triangular matrix, in which the rows and columns represent the sentences of the text.

$$\text{Text level argument overlap} = \sum_{i=1}^{n} \sum_{j=1}^{n} P_{ij}|i < j * \frac{1}{n * \frac{n-1}{2}}$$

*Semantic overlap* indicates the semantic similarity between sentences. To measure the semantic overlap we applied the common statistical procedure latent semantic analysis (LSA) as introduced by Graesser et al. [46]. LSA values can be interpreted as a correlation that takes values between 0 (low semantic relatedness) and 1 (high semantic relatedness). LSA is a sensitive feature for the semantic relatedness of sentence pairs and is a central indicator of text cohesion in Coh-Metrix [48]. The LSA-cosine is computed as follows (see Fig 3): First, LSA counts how often each word occurs in each sentence. Then, a statistical procedure called singular value decomposition is applied which reduces the number of words to a smaller number of main concepts. From this reduced form, the semantic similarity between sentences is then computed by taking their cosine similarity. Values close to 1 indicate a high semantic similarity and values close to 0 indicate a low semantic similarity. Adjacent semantic overlap measures the cosine similarity between neighboring sentences, and text level semantic overlap measures the cosine similarity between all possible sentence pairs of a text.

*Divergent features on the syntactic level.* Average length of longest dependency refers to the longest distance between a word and its dependent within a sentence [49]. For example, the sentence "The tall and energetic politician who emphasized the value of freedom did not like the reporter" (see Fig 3) has a longest dependency of 10: The subject "politician" must be related to the predicate "like" [50] and the distance between the subject and the predicate is 10 words. Long dependencies are challenging for readers, as the first element of the dependency has to be kept in working memory until the next element can be related to the first element [50]. Thus, the average length of the longest dependency for each sentence in a text is a precise feature for the syntactic complexity of a text.

*Average number of complex nominals per clause* refers to groups of words that relate to a common noun [51]. For example, a noun can be described in more detail by one or more adjectives (e.g., "the tall and energetic politician"), by placing multiple noun phrases side by side (e.g., "*the man who* drank beer"), or by a gerund phrase (e.g., "*doing sports regularly* is good for your health"). To obtain the average number of complex nominals per clause, the number of complex nominals is counted and then averaged by the number of clauses. Complex nominals are challenging for readers since they put a high demand on working memory, especially for less proficient readers [52].

*Divergent features on the lexical level. Word concreteness.* Concrete words are characterized by the fact that they refer to tangible sensory experiences. By using concrete words, a reader can see agents performing actions that affect objects. For example, the word "reporter" refers to an object and a sensory experience, whereas the word "freedom" refers to a concept that cannot be captured by sensory experiences and is thus more abstract. To measure word concreteness, several computational models have been proposed [53]. Among those, word concreteness models based on WordNets are commonly used. A WordNet is a lexical database of words from a particular language that form a hierarchical tree structure. The closer words are to the root of the tree structure, the more abstract they are. For example, the word "sports" is

related to "soccer" since "sports" is a superordinate of soccer. Hence, "sports" has fewer hypernyms (i.e., superordinates) than "soccer" and is thus more abstract than "soccer". To compute the hypernym score, we calculated the average number of hypernyms among all concepts within a text. To compute these scores, we used GermaNet, a WordNet for the German language containing over 16,000 words [54]. High values in the hypernym score signify high levels of concreteness. Low levels of the hypernym score characterize low levels of concreteness.

*Root type-token ratios* are a common feature of lexical diversity. Type-token ratios are calculated by dividing the number of unique words (types) within a text by all words within a text (tokens). For example, the sentence in Fig 3 consists of 13 words (tokens) and 12 unique words (types), since politician is repeated once in the second sentence and stop words such as articles are ignored. Hence, this example would yield a type-token ratio of 12/13 or 0.92. As type-token ratios are sensitive to text length, root type-token ratios are adjusted for text length by taking the square root of the number of tokens [55]. Texts with a high root type-token ratio are more difficult to understand because readers need a greater vocabulary to understand the content of a text.

## Data analysis

We tested the reliability and the validity of the CohViz concept maps using the two corpora (see Fig 4). All texts from both corpora were automatically analyzed by the CohViz algorithm. The algorithm generated the concept maps from the texts of the respective corpus and computed the three structural indicators from these concept maps (i.e., number of fragments, number of concepts, number of relations). These structural indicators were the basis for testing the reliability and validity of the feedback.

**Assessing the reliability of CohViz.**  To test the reliability of CohViz, we first created the reliability corpus. These texts were fed into the CohViz engine and additionally analyzed by human expert raters who segmented each text into propositions. From these propositions we computed the concept maps and extracted their structural indicators. To compare the three structural indicators between the CohViz concept maps and the human expert concept maps, we computed product-moment correlations, interrater agreement, and bias scores. Interrater agreement was measured by intraclass correlations. We additionally chose to use intraclass correlations to product-moment correlations since product-moment correlations do not take potential differences among raters into account and only provide evidence for the general associations among two raters. Therefore, generally intra-class correlations are preferred, as they additionally correct for potential divergence among two or more raters [56]. Interrater reliability measures, however, only measure the (dis)agreement between the computer feedback and the human expert raters but do not provide additional information about the direction of potential disagreements (i.e., over- or underestimations) [57]. The direction of disagreement is commonly computed in terms of bias measures [58]. Bias refers to the signed difference between the values provided by the computer-based feedback tool and the values by the human expert raters ($X_{Computer} - X_{rater}$). Positive values indicate overestimations (i.e. the computer provides higher numbers than the expert raters), negative values indicate underestimations (i.e. the computer provides lower numbers than the expert raters). These general quantitative methods are often accompanied by qualitative analyses of the disagreements to inspect the underlying reasons for these disagreements [6,59,60]. We used an alpha level of .05 for all statistical analyses. Power analyses indicated that we achieved an excellent test-power of $1-\beta = .93$ (while setting $\alpha$-error to .05, the sample size to $N = 100$, and the smallest detectable effect to $r = .30$).

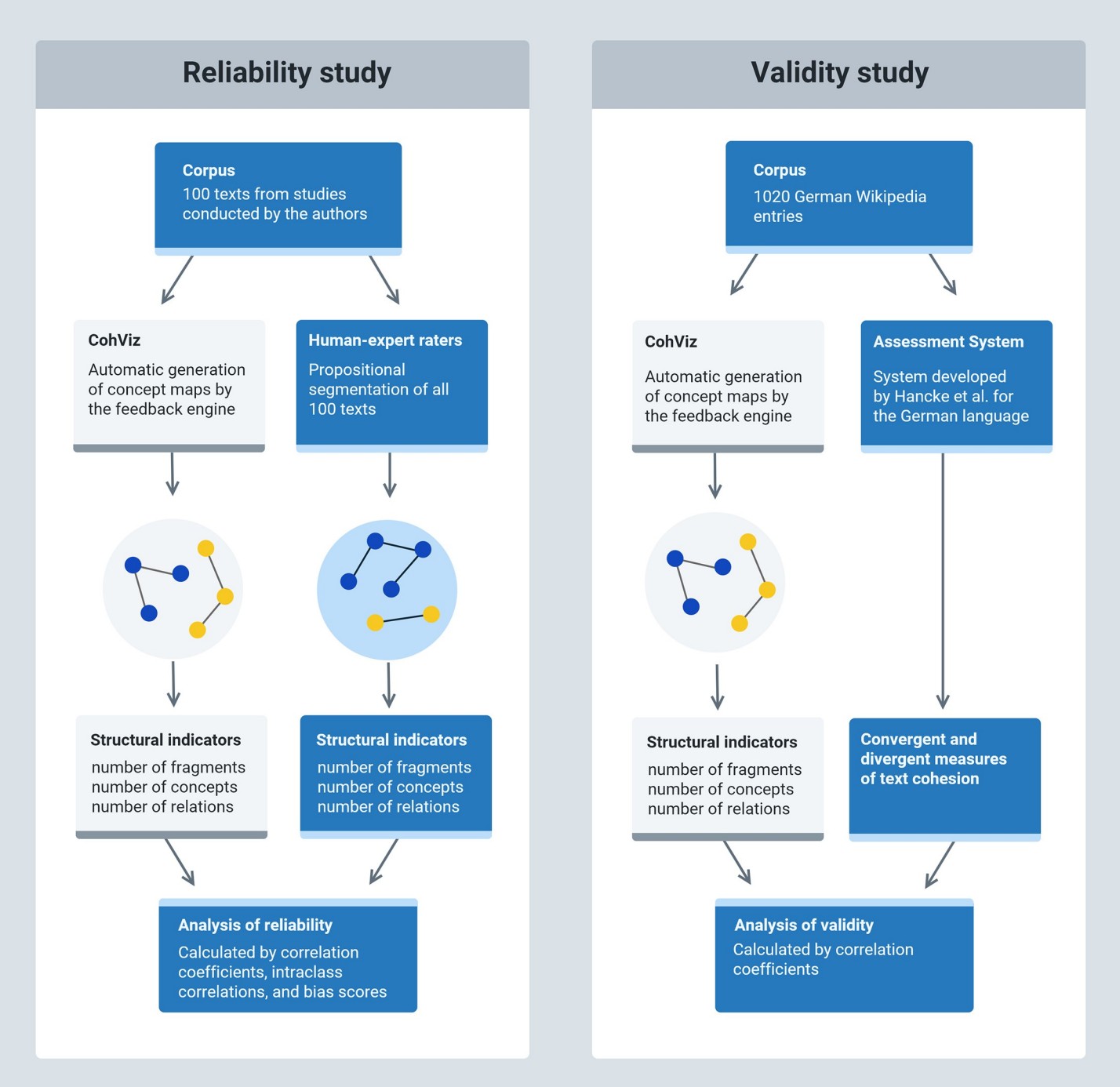

**Fig 4. Depiction of the data analysis procedure for the reliability and validity study.**

**Assessing the validity of CohViz.** To test the validity of CohViz, we proceeded similarly. First, we fed the texts from the validity corpus into the CohViz engine which extracted the three structural indicators. Next, we computed the convergent and divergent measures of text cohesion from the texts of the validity corpus with the assessment system developed by Hancke et al. [37]. To compare the three structural indicators generated by CohViz with convergent

and divergent measures of cohesion, we computed product-moment-correlations in particular for the number of fragments as the central measure of text cohesion delivered by CohViz. Post-hoc analyses indicated that we achieved an excellent test-power of $1-\beta = .94$ (while setting $\alpha$-error to .05, the sample size to $N = 1020$, and the significantly smallest detectable effect to $r = .01$). As such, for the interpretation of the convergent and divergent validity of CohViz, the direction and the size of the effect (i.e., the size of the correlation) is more important than the mere establishment of significance, as small but less meaningful correlations may also become significant due to the high test power.

## Results

### Reliability

First, to investigate the relationships between the concept maps generated by CohViz and the human expert raters, we computed product-moment correlations and intraclass correlations (*ICC*) per dependent measure (see Table 3). For the number of fragments, we found high correlations between the CohViz concept maps and the concept maps based on human expert raters. Similarly, the intraclass correlations for the number of fragments were high. For the number of relations, we obtained a high correlation between the maps generated by CohViz and the maps based on human raters, and medium-to-high intraclass correlations. The correlation for the number of concepts in the CohViz-generated and human rater based maps was excellent, as was the intraclass correlations (see Table 3). Together, these results indicate that concept maps generated by CohViz were generally highly consistent with the concept maps by human raters, as the product-moment and intra-class correlations showed at least medium but mostly high values for the three measures, that is, the number of fragments, the number of concepts, and the number of relations.

To investigate the direction of potential discrepancies, we computed bias scores [58] between the human expert concept maps and the CohViz concept maps for each dependent variable (i.e., the number of fragments, the number of concepts, the number of relations, see Table 3). To test whether these differences were significant, we computed one-sample *t*-tests against zero for each dependent variable [58]. As the correlations already indicated, we did not find a significant difference in the number of fragments between the human expert raters and CohViz ($d = 0.15$, small effect), indicating that CohViz was highly accurate in detecting cohesion fragments in the text corpus. However, CohViz tended to overestimate the number of relations ($d = 1.39$, large effect), and the number of concepts ($d = 0.67$, medium effect), indicating lower accuracy regarding the extraction of concepts and relations from the texts.

**Table 3. Accuracy of the CohViz methodology compared to human raters.**

| Feature | Human raters | | CohViz | | Product-moment correlation | | Intra-class reliability | Bias scores | | | | 95% CI | | Size of effect |
|---|---|---|---|---|---|---|---|---|---|---|---|---|---|---|
| | *M* | *SD* | *M* | *SD* | *r*(98) | *p* | *ICC* | $M_{diff}$ | $SD_{diff}$ | *t*(99) | *p* | *LL* | *UL* | Cohen's *d* |
| Number of fragments | 2.63 | 1.71 | 2.76 | 1.79 | .76 | < .001 | .76 | -0.13 | 1.22 | -1.07 | .289 | -0.37 | 0.11 | -0.15 |
| Number of concepts | 30.90 | 12.13 | 32.63 | 12.32 | .83 | < .001 | .78 | -1.73 | 3.64 | -4.75 | < .000 | -2.45 | -1.01 | -0.67 |
| Number of relations | 40.61 | 20.98 | 59.21 | 29.90 | .95 | < .001 | .96 | -18.60 | 17.00 | -10.93 | < .000 | -15.18 | -21.92 | -1.39 |

$M_{diff}$ = mean difference scores between human raters and CohViz; $SD_{diff}$ = standard deviation of difference scores between human raters and CohViz; *ICC* = intraclass correlations; *CI* = confidence intervals; *LL* = lower limit, *UL* = upper limit. All t-tests were two-tailed.

To investigate the reasons for these overestimations regarding the extraction of concepts and relations, we conducted two follow-up qualitative analyses.

First, we analyzed the overestimation in the number of concepts for the 10 texts with the largest number of overestimations regarding the number of concepts. On average, the selected subsample of CohViz concept maps contained 10 concepts more than the human expert raters' concept maps (which corresponds to an overall overestimation of 101 concepts for all 10 of the texts). Of these overestimations, 22% ($n = 22$) occurred due to parsing errors of the CohViz engine during the extraction phase (see Fig 2, e.g. by wrong categorizations of n-grams, or lemmas). The remaining 78% of the overestimations ($n = 79$) occurred because raters accidentally omitted concepts or entire sentences in their propositional segmentations. Thus, the qualitative analysis rather suggests that the disagreement between CohViz and the human expert raters regarding the number of concepts resulted because of segmentation errors by the human expert raters and not by the CohViz system, which further corroborates the reliability of the CohViz system.

We proceeded similarly for the number of relations and analyzed the overestimations of the ten texts with the highest overestimations of relations. On average, each concept map by CohViz contained 47 relations more than the human expert raters' concept maps (which corresponds to an overall overestimation of 476 relations for all 10 texts). Of these overestimations, 66% ($n = 315$) were due to errors of the CohViz system. The qualitative analyses revealed that most overestimations resulted from texts that contained complex sentence structures (e.g., long sentences comprising several subordinate clauses), or realized enumerations in the form of incomplete sentences. Thus, most of the errors must be attributed to parsing problems in the extraction phase, as computer-linguistic approaches usually require well-formed grammatical constructions. Contrarily, 34% ($n = 161$) of the potential overestimations resulted due to omissions of the human raters, which took place during the segmentation of the students' texts. Thus, the qualitative analysis suggests that the overestimations regarding the relations among concepts likely resulted when the text contained relatively complex syntactic structures and incomplete sentences.

## Validity

**Correlations of CohViz with convergent features of text cohesion.** We computed product-moment correlations to investigate the relationship between CohViz features, in particular, the number of fragments as the central measure of text cohesion delivered by CohViz, and the different convergent features. In line with our expectations, we found medium to large negative correlations for most of the convergent features of text cohesion (except for adjacent semantic overlap, see Table 4) with the CohViz indicator number of fragments, indicating that an increase in CohViz fragments was associated with a decrease in cohesion. The text level measures of cohesion correlated more strongly with the CohViz measures than the adjacent cohesion measures, as the number of fragments is rather a measure of cohesion gaps on the entire text level.

**Correlations with divergent features of text cohesion.** Next, we tested associations of the number of fragments provided by CohViz with the divergent measures of writing quality (i.e., syntactic level and lexical level). Except for the average length of longest dependencies, we found non-substantial to weak correlations with the number of fragments generated by CohViz for both the syntactic and the lexical features (see Table 4). One reason for the particularly weak correlation between the average number of longest dependencies and the number of fragments might be that CohViz generates more relations with an increase in the average length of dependencies, leading to a decrease in the number of fragments. In summary, these

**Table 4. Correlations between CohViz features and convergent and divergent features of text cohesion.**

| Level | | Feature | 1 | 2 | 3 | 4 | 5 | 6 | 7 | 8 | 9 | 10 | 11 |
|---|---|---|---|---|---|---|---|---|---|---|---|---|---|
| CohViz features | | | | | | | | | | | | | |
| | 1. | Number of fragments | – | | | | | | | | | | |
| | 2. | Number of concepts | .12*** | – | | | | | | | | | |
| | 3. | Number of relations | -.04 | .57*** | – | | | | | | | | |
| Convergent features on the cohesion level | | | | | | | | | | | | | |
| | 4. | Adjacent semantic overlap | **-.25***** | -.02 | .10*** | – | | | | | | | |
| | 5. | Text level semantic overlap | **-.43***** | -.11*** | .10** | .66*** | – | | | | | | |
| | 6. | Adjacent argument overlap | **-.46***** | -.00 | .09** | .40*** | .32*** | – | | | | | |
| | 7. | Text level argument overlap | **-.60***** | -.06 | .09** | .25*** | .44*** | .75*** | – | | | | |
| Divergent features on the syntactic level | | | | | | | | | | | | | |
| | 8. | Average length of longest dependency | **-.24***** | .45*** | .29*** | .17*** | .27*** | .21*** | .28*** | – | | | |
| | 9. | Average number of complex nominals per clause | **-.08**** | .23*** | .19*** | .12*** | .20*** | .06 | .07* | .29*** | – | | |
| Divergent features on the lexical level | | | | | | | | | | | | | |
| | 10. | Word concreteness | **-.02** | -.25*** | -.22*** | -.03 | -.03 | .01 | .02 | .07* | -.30*** | – | |
| | 11. | Root type-token ratio | **.15***** | **.81***** | .36*** | -.06 | -.15*** | -.12*** | -.17*** | .52*** | .15 | .08* | – |

The central results from the validity study are in boldface. Correlations represent Pearson product-moment correlation coefficient

*p ≤ .05

**p ≤ .01

***p ≤ .001

results indicate that the number of fragments was not substantially related to most divergent features of writing quality. These findings on the discriminant features further underscore the conclusion that CohViz is a valid indicator of text cohesion.

## Discussion

The purpose of the presented studies was to examine the reliability and validity of CohViz, a computer-based feedback approach, which provides (novice) writers with informative feed-back on the cohesion of their writing in the form of concept maps. We successfully demon-strated the reliability and construct validity of the CohViz feedback in two studies.

The findings of the reliability study showed that CohViz is particularly reliable in visualiz-ing cohesion deficits and concepts. However, regarding the depiction of relations, CohViz tended to be less accurate compared to the human expert raters, indicating room for technical improvement. Regarding the number of concepts, our qualitative analyses revealed that Coh-Viz proved to even be superior to human expert raters in validly visualizing the text's concepts, as human coding was often accompanied by coding errors. This finding is astonishing, as both expert raters received in-depth training of several hours on propositional segmentation and the instantiation of cohesion. Apparently, computer-generated concept maps by the CohViz engine were less error-prone than human-generated concept maps and as such may offer valid opportunities to provide students with accurate feedback on the chosen concepts when writing.

In the validity study, we demonstrated the convergent and divergent validity of the CohViz system using features of writing quality from well-established assessment systems [36,46], as used by Coh-Metrix [46]. In particular, we compared the fragments as the main indicator of text cohesion in CohViz with convergent features of text cohesion (i.e., argument overlap; semantic overlap) and divergent features on the syntactical and lexical level (i.e., average length

of longest dependency; average number of complex nominals per clause; word concreteness; root type-token ratio). We found that the number of fragments was moderate to highly associated with other convergent features of text cohesion. In line with our expectation of divergent validity, there were no substantial relations between the number of CohViz fragments and the lexical and syntactical features. The results from the validation study suggest that the fragments as the central indicator of text cohesion are particularly valid to assess the degree of argument overlap in students' texts. In particular, the fragments seem to assess how connected the central concepts are on the text level. This result is particularly interesting as it highlights the gap in the mechanisms that contribute to the effectiveness of feedback. Given this result, it can be assumed that the fragments will help students, in particular, to identify issues of textual cohesion at the level of argument overlap.

The two studies have both theoretical and practical implications. First, our findings add to the scarce evidence that human expert raters, as the presumed gold-standard, are not always a better alternative in segmenting and parsing texts compared to computer-based processing technology. In our study, we recruited expert raters with ample training on propositional segmentation. Nevertheless, these expert raters still faced difficulties in accurately segmenting expository texts into concepts. On the other hand, expert raters were better able to detect relations among concepts, as human expert raters have a more profound knowledge of grammar and linguistics. Our findings thus show that human expert raters are not necessarily the gold-standard to analyze texts in terms of its structural indicators. The superiority of computer-based approaches over human raters has already been suggested for surface writing tasks such as spelling or orthography [61]. Our findings, however, add to the previous findings, as they show that computer-based systems may be suited to assess the cohesion of texts, which can be regarded as an important feature that contributes to the quality of texts [11,62]. Thus, we see computer-based assessments as a potential supplement to human assessments to further advance students' writing [12].

Finally, our study highlights the central role of triangulation in validation studies. Many validation studies often solely relied on reporting relationships with other assessment technologies or simply reported relations to human expert raters [63–66]. The findings of these studies provide relevant insights into potential relationships among tools and human raters. Such uni-dimensional analyses, however, may not suffice to fully prove the reliability and validity of the features under investigation, as no information is available about the direction of potential discrepancies and the underlying reasons. In our studies, by contrast, we implemented multiple quantitative features to examine the overall validity of the CohViz feedback. These analyses were complemented by qualitative analyses, which additionally exemplified potential reasons for the inaccuracies. As such, we see the triangulation method in our studies as a valuable approach to investigate the validity of computer-based assessments.

## Limitations of the study and future research

Since the results of the current studies suggest that the fragments in CohViz are indicative of problems in argument overlap at the text level, future research should conduct empirical studies to test whether indeed argument overlap is best supported by the feedback. Since the ability to establish cohesion by argument overlap typically develops at a younger age [67], future studies should investigate the effectiveness of the feedback with these students. Such studies would not only provide crucial information for which aspects of cohesion the tool is most effective but also increase the generalizability of the feedback.

Since we only used expository texts (i.e., explanatory texts) in our studies it is not clear whether our findings would generalize to other text genres such as narrative texts. For

instance, in narrative texts, text cohesion is lower than in expository texts [8,68]. That said, genres may not only differ in the extent of text cohesion but may also rely on different linguistic devices to instantiate cohesion within texts. For instance, in expository texts, experienced writers rather use argument overlap and bridging information as linguistic devices, whereas in narratives, cohesion is often achieved by the use of pronominal references. Given that the instantiation of cohesion may highly depend on the particular genre, future studies should, therefore, replicate our findings in other genres.

Another related issue refers to the text length in our studies (around 200–300 words). Given that particularly semantic features based on latent semantic analyses work better for longer texts [6], the question remains whether our findings would replicate with longer texts. Then, however, CohViz concept maps might become too large and complex. One solution could be to provide students with several small and manageable concept maps portraying the cohesion for the different subsections of the text. A less detailed but more inclusive "macro-concept map" could visualize the central theme of the text by showing global relations between text elements. Whether novice writers will benefit from feedback consisting of such a combination of concept maps that differ in the level of detail is also a question to be addressed in further studies.

## Conclusion

In conclusion, the two studies showed that CohViz is a reliable and valid approach to provide students with feedback on the cohesion of their writing. Verifying the accuracy of the feedback is an important step to uncover the mechanisms underlying the effectiveness of concept map for text cohesion. The findings of the two studies provide evidence that CohViz is an effective tool to improve the cohesion of students' texts *because* it accurately depicts deficits of text cohesion. Thus, CohViz may serve as a vehicle to supplement instructors' feedback and enhance the cohesion of students' writing.

## Acknowledgments

We would like to thank Christina Schuba, Ricarda Budde, Arne Kappis, and Helen Dambach for coding the corpus from the reliability study. We furthermore express our gratitude to Zarah Weiß and Detmar Meurers from the Department of Theoretical Computer Linguistics (ISCL) of the University of Tübingen for providing the automated measures in the validity study.

## Author Contributions

**Methodology:** Andreas Lachner.

**Software:** Christian Burkhart.

**Supervision:** Matthias Nückles.

**Writing – original draft:** Christian Burkhart.

**Writing – review & editing:** Andreas Lachner.

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
