## [Decision Letter · Decision Letter 0]

24 Jan 2020

PONE-D-19-33124

Assisting students’ writing with computer-based concept map feedback: A validation study of the CohViz feedback system

PLOS ONE

Dear Mr. Burkhart,

Thank you for submitting your manuscript to PLOS ONE. After careful consideration, we feel that it has merit but does not fully meet PLOS ONE’s publication criteria as it currently stands. Therefore, we invite you to submit a revised version of the manuscript that addresses the points raised during the review process.

In particular:

Some of the the analysed measures should be defined formally (e.g. "mean readability score based on the Flesch-Kincaid measure"),Details of statistical analysis procedure should be given,Demographic information of students and human experts is missing,The information about the authors of CohViz system should be clearly presented,The text should be changed to remove not needed repetitions of sentences and information.

We would appreciate receiving your revised manuscript by Mar 09 2020 11:59PM. To enhance the reproducibility of your results, we recommend that if applicable you deposit your laboratory protocols in protocols.io, where a protocol can be assigned its own identifier (DOI) such that it can be cited independently in the future. For instructions see: http://journals.plos.org/plosone/s/submission-guidelines#loc-laboratory-protocols

We look forward to receiving your revised manuscript.

Kind regards,

Maciej Huk, Ph.D.

Academic Editor

PLOS ONE

Journal Requirements:

2. Please provide additional details regarding participant consent. In the ethics statement in the Methods and online submission information, please ensure that you have specified whether consent was informed.

Reviewers' comments:

Reviewer's Responses to Questions

**Comments to the Author**

1. Is the manuscript technically sound, and do the data support the conclusions?

Reviewer #1: Yes

Reviewer #2: Yes

Reviewer #3: Partly

2. Has the statistical analysis been performed appropriately and rigorously? 

Reviewer #1: I Don't Know

Reviewer #2: Yes

Reviewer #3: I Don't Know

3. Have the authors made all data underlying the findings in their manuscript fully available?

Reviewer #1: Yes

Reviewer #2: Yes

Reviewer #3: Yes

4. Is the manuscript presented in an intelligible fashion and written in standard English?

Reviewer #1: Yes

Reviewer #2: Yes

Reviewer #3: Yes

5. Review Comments to the Author

Reviewer #1: I think the quality and overall rigor of the studies are very good; however, the writing about the statistics in Study 2 is confusing enough that I can't assert that the work meets the Plos One standard. (Although I do think it likely does.)

I also attached the below comments in a separate document, but overall, I do think this is a publishable paper. I would like to see more practical advice for using the tool.

Major comments:

This is a helpful paper because it validates the use of a specific tool for student feedback on their writing. It is unclear how the authors think this tool is being used and how it can be used.

The paper is very long and some information seems unnecessary. There are, for example, many examples. There is a lot of repetition both on a sentence level and in terms of information repeated again in later explanations that should be adjusted.

The paper is flawed as a piece of writing because of the way that the two studies were basically pasted together as well as the chronological presentation of information. The paper needs to be reorganized to explain that the authors studied validity and reliability of the CohViz system and have one methods section that outlines all the methods (preplanned and post hoc) and one results and discussion section. The paper does need to be rewritten and some of the framing and examples have to be sacrificed to make the paper work as a whole.

The current overall discussion section is very good—really concise, helpful, and well-written. The rest of the paper should follow this model and state things once and without so much explanation.

The results should be tabulated and not embedded in the text.

Specific comments:

Abstract:

• Overall this is clear and well written, describing a helpful pair of studies about a tool already in use.

• It would be helpful to know the discipline of the teachers using CohViz

• The authors should be careful not to use the same word multiple times in a sentence or phrase

• It would be helpful to know the general length of the texts assessed

Introduction:

• Please delete the information about aspirin and the following sentence. Your prior point is cogent and adequate and does not need more support.

• It would be helpful to have a little more information about the setting for the studies. Just a couple of words—college? High school? English? Writing? Wikipedia?

• Overall—the introduction does not seem to me to be introducing the major concepts that occur later in the paper and therefore it is confusing

Computer-Based Systems on Students’ Cohesive Writing

• I’m not sure what this section adds. Would it be sufficient to have a single paragraph providing a short overview of the idea of using a computer-based system for evaluating cohesion in the earlier section?

CohViz: A Feedback System to Support Students’ Cohesive Writing

• I still would like more information about the setting for using this tool—who uses it?

• I am unsure when and where the cited studies by Lachner and colleagues were completed. In a college? Recently? Why?

Accuracy as a Critical Component of Effective Computer-Based Feedback, Reliability, Validity

• I’m not sure what these sections add. Ideally, the section about CohViz should simply state the limitations of the prior studies, the actual needs of users, and the need for the current studies much more concisely, following the style of the current overall discussion.

Overview of the Present Studies

• I would not describe the studies as “empirical”

• I would describe the studies in terms of what they are testing (“reliability” and “validity”) instead of calling them Study 1 and Study 2 and expecting the reader to keep track of the shifts

Study 1 Methods:

• I find this section confusing, largely because I’m not sure who is doing what where and when and to which corpus of texts. Ideally the methods would identify in order:

o The goal of the study (assess reliability of concept maps between machine and human coders)

o How human coders were identified

o Which corpus of work was used for CohViz versus human coders. How was the full corpus generated and in which setting and by whom? How was the smaller representative sample generated in which setting and why whom?

o How were the outputs generated? Who reviewed them?

• I would move the explanation about how CohViz works to the introduction

• I would strongly recommend avoiding the use of “Lincoln freed the slaves” as an example. It would be highly offensive to many US readers. Please choose something more neutral. Maybe “Hollywood baked the rolls”?

Study 1: Results and discussion:

• Please move the statistical methods back to the methods section

• This discussion is far too long and has far too many little asides and explanations As a reader, it is helpful to have the results presented as:

o How big were the samples reviewed? Did the representative sample for human coders adequately mimic the overall sample? How did you know that?

o What actually was completed? How many concept maps were generated and reviewed?

o How well did the two methods correlate?

Study 2:

• This seems very long and not clear because information that should be in the introduction is interspersed throughout the text, there are methods in the results section and too much explanation as the results are presented. It simply feels very similar to the problems I noted in Study 1, where the methods and results are not clearly laid out in a logical order, but rather seem to demand that the reader carefully track with the researchers.

Overall discussion:

This is generally well written and concise. Please describe the studies in terms of what they tested and not by numbers. It makes the work very hard to read.

Limitations and future directions:

This seems very wordy and a bit too speculative. I applaud the correctness of the limitations identified, but the lengthy caveats and explanations are not really helpful. More helpful are the practical modes of moving forward.

Reviewer #2: The authors showed that CohViz is a liable and valid approach to provide students with feedback on the cohesion of their writing. A major cause which might lead to an almost-reject decision is that their research purpose is not convincing. Specifically, the authors claim at the beginning that the effectiveness of CohViz may come from a superficial warning or an essential improvement on cohesion. And they find at last that the effectiveness of CohViz comes from the essential improvement on cohesion. The reason behind their questioning that the effectiveness of CohViz may only come from a superficial warning is not solid. Generally speaking, CohViz-like systems are just developed from the idea that bad writing usually companies with bad cohesion. It is hard to believe that these systems do not benefit from the above design idea. As in the Aspirin example illustrated by authors themselves, the authors’ work in CohViz is like to proof that Aspirin is not a placebo. I believe this is not enough, in terms of innovation. If the authors can provide an example telling that some students do not actually improve cohesion suggested by CohViz-like systems, but make other changes which also improves the quality of writing, I will be more than happy to recommend their revisions to be accepted. Other problems are listed as follows.

1) Second Paragraph in Introduction:

The first question is: Which references among [4,5,10,11] do the authors believe to document CohViz is a beneficial tool? I believe there is none, otherwise they ought to provide a specific reference indicating where Figure 1 comes from. I suggest that the authors first define CohViz as a family/cluster of computer-based feedback systems which help students’ writing using cohesion. If CohViz is really a specific tool, then the success of W-pal (the real system developed in [10]) does not necessarily mean CohViz is also a success. At least, the authors may add that CohViz is developed based on a duplication of W-pal.

The second question is: I understand the purpose of authors using the Aspirin example, but it is far from CohViz. The authors may simply indicate that they know CohViz is effective, but do not know how it works or do not know the detail mechanism makes CohViz effective. That is enough. NO need to use an irrelevant example to point out the logic error (even do not need to mention the logic error because it may probably be the authors’ own logic, not others’). Other readers may simply want to know more, e.g., why is it effective? If this is true, then there is actually no logic error. Please go quick and straight to the point. And I am also disappointed when reading the conclusion of this manuscript that it only confirms CohViz’s effectiveness, but seemingly forgets the why question.

To sum up, the 2nd paragraph in Introduction needs to be re-written.

2) Page 5, the first sentence under the subtitle “CohViz: a feedback system to support students’ cohesive writing”. I did not find any “CohViz” in [5]. Do you mean [5] developed “CohViz”? I believe not.

3) Page 6, Lachner, Burkhart, and Nückles are not the authors of [5], and they are also not the authors of [4].

4) Related work is not reviewed adequately. Other works can be used as a start point, such as [5]. The authors may tell readers what [5] has already achieved in the mechanism investigation.

5) The common senses of reliability and validity do not need to be re-introduced.

6) The demographic information of students and human experts is missing.

Reviewer #3: >>> 1. Language problems:

1.1 hypernyms => hyperonyms (many times)

1.2 funders => founders

>>> 2. Presentation problems:

2.1 Table 1: the table title nor its header do not specify the meaning of values in brackets

2.2. The format of references is not uniform (others => et al.)

2.3 Fig 1., Fig 3., the quality is low, please consider vector format

>>> 3. Other problems:

3.1 some of the the analyzed measures were not defined formally, e.g. "mean readability score based on the Flesch-Kincaid measure",

"Adjacent semantic overlap", "Word concreteness"

e.g. "Adjacent semantic overlap measures the cosine similarity between neighboring sentences, and text level semantic overlap measures the cosine similarity between all possible sentence pairs of the text." is not explaining how the measure is calculated.

3.2 Obtained results can be biased by the selection of the human experts. The process of their selection is not given.

3.3. It would be good to give information who is the author of CohViz system. In the actual form of the text it is unclear.

Summary: The language is good. The quality of illustrations is low and the presentation should be improved. But what is most important area that needs fixing includes definitions of analyzed measures and analysis of the characteristics of used population of human experts.

Recommendation: major rework

6. PLOS authors have the option to publish the peer review history of their article (what does this mean?). If published, this will include your full peer review and any attached files.

Reviewer #1: Yes: L DeTora

Reviewer #2: No

Reviewer #3: No

---

## [Author Response · Author response to Decision Letter 0]

9 May 2020

Reviewer #1: 

I think the quality and overall rigor of the studies are very good; however, the writing about the statistics in Study 2 is confusing enough that I can't assert that the work meets the Plos One standard. (Although I do think it likely does.)

I also attached the below comments in a separate document, but overall, I do think this is a publishable paper. I would like to see more practical advice for using the tool.

Major comments:

#1: This is a helpful paper because it validates the use of a specific tool for student feedback on their writing. It is unclear how the authors think this tool is being used and how it can be used.

In the introduction of the revised manuscript we added a section in which we explain how and for what purposes the tool can be used (i.e., “CohViz: A feedback system to support students’ cohesive writing”, see p. 4). We argue that the main advantage of CohViz is that it provides students with instant feedback on the cohesion of their texts and thus supplements teachers‘ feedback capabilities. In particular, we provide users with use cases in which we explain how teachers can use the feedback as a modeling tool and how students can revise their texts, for example in terms of a homework assignment. In addition, in the discussion, we now explain how the results of the current study can contribute to the utility of the tool. In that section we argue that the results have shown that CohViz is particularly valid for deficits in students’ ability to establish argument overlap. As a practical recommendation we therefore suggest that in future studies the effectiveness of the tool should be tested with writers who have specific problems with argument overlap.

#2: The paper is very long and some information seems unnecessary. There are, for example, many examples. There is a lot of repetition both on a sentence level and in terms of information repeated again in later explanations that should be adjusted.

We thoroughly went over the whole manuscript and have tried to remove all repetitions. In the course of restructuring the manuscript (see #3) we additionally removed many repetitons on the overall structure of the manuscript.

#3: The paper is flawed as a piece of writing because of the way that the two studies were basically pasted together as well as the chronological presentation of information. The paper needs to be reorganized to explain that the authors studied validity and reliability of the CohViz system and have one methods section that outlines all the methods (preplanned and post hoc) and one results and discussion section. The paper does need to be rewritten and some of the framing and examples have to be sacrificed to make the paper work as a whole.

We have thoroughly reorganized the entire manuscript. The revised manuscript now has only one methods, results, and discussion section. In the introduction we focused more on the limitations from previous studies and the need to study the accuracy of CohViz. In the method section we followed the rhetorical structure of similar studies published in PlosOne and divided the methods section in the subsections “Corpora collection”, “Measures”, and “Data Analysis” (see Abu-Shaheen et al., 2018; Hara et al., 2016; Kogure et al., 2014; Kruizinga et al., 2012; Steenson et al., 2018). In the results section we again followed the outline of previous studies and separately presented the results for the reliability and validity of the feedback. In addition, as mentioned in #2, we removed unnecessary repetitions and examples from the manuscript.

#4: The current overall discussion section is very good—really concise, helpful, and well-written. The rest of the paper should follow this model and state things once and without so much explanation.

As of comment #1, we have tried to follow the style of the overall discussions and have tried to remove repetitions from the manuscript.

#5: The results should be tabulated and not embedded in the text.

We went through the results carefully and looked for data that was not reported in tabular form. We found one case in Study 1 (now the reliability study) and integrated these results into Table 3. In addition, we have moved information about the two corpora presented in the body text to their corresponding tables.

Specific comments:

#6: Abstract: Overall this is clear and well written, describing a helpful pair of studies about a tool already in use. It would be helpful to know the discipline of the teachers using CohViz.

So far, CohViz has mainly been used for research purposes in laboratory settings and in ecological valid classroom situations (e.g., philosophy education, educational research, teacher education). On page 6, we have included the information accordingly: 

"So far, the effectiveness of CohViz has been tested in various settings between 2017 and 2019, including controlled laboratory studies and ecologically valid field studies in a wide range of disciplines (e.g. biology, ethics, philosophy, educational psychology, teacher education). In these experimental studies, the authors examined the effectiveness of CohViz with college students [15,20,21,29]. Overall, in a mini meta-analysis Burkhart, Lachner, and Nückles [30] could show that CohViz has a medium effect on both local and global cohesion and was thus effective in improving the cohesion of students’ texts (i.e., local cohesion g = 0.62; global cohesion g = 0.57). A potential explanation for the obtained effects of CohViz can be found in the think-aloud study by Lachner, Burkhart, Nückles [15]. The think-aloud study could show that students who processed the concept maps could infer both local and global writing plans from the concept maps directly. In addition, the analysis of the concept map triggered negative monitoring processes (i.e., student thought about the macrostructure of a text) which lead to further planning processes."

#7: The authors should be careful not to use the same word multiple times in a sentence or phrase

As mentioned in comment #1 and #4, we have carefully revised the manuscript for potential word reduncancies and deleted them accordingly.

#8: It would be helpful to know the general length of the texts assessed.

The length of the texts of the two corpora are reported in Table 1 and Table 2. In the revised manuscript we also moved some descriptions of the corpus (e.g., number of sentences) from the body text to the tables. For both studies we now report the number of words and sentences in their respective tables (see page 10 and 12).

#9: Introduction: Please delete the information about aspirin and the following sentence. Your prior point is cogent and adequate and does not need more support.

We removed the information about aspirin from the manuscript.

#10: It would be helpful to have a little more information about the setting for the studies. Just a couple of words—college? High school? English? Writing? Wikipedia?

In the revised manuscript, we added some information about the setting of the studies (see p. 8): 

„The corpus to test the reliability of CohViz was compiled by the authors as a sample from an entire set of 901 expository texts written in German by college students. All texts were produced by novice writers in the course of different experimental studies conducted by the authors and were conducted between 2015 and 2018 at German Universities. We made sure to compile texts with a representative range of topics in the natural sciences and the humanities. In all of these studies the dependent variables of interest were measures of text quality (e.g., local and global cohesion).”

#11: Overall—the introduction does not seem to me to be introducing the major concepts that occur later in the paper and therefore it is confusing.

We have thoroughly restructured the introduction to make the major concepts of the manuscript clearer. After providing the reader with an introduction to the lack of research of the feedback’s accuracy, we introduce CohViz as the main computer-based feedback system of this manuscript (see heading “CohViz: A feedback system to support students‘ cohesive writing”). Next, we explain that previous research only measured the external validity of the tool and introduce the reader to the lack of research on the of reliability and validity of the feedback (see heading “Previous research on the effectiveness of CohViz”). We then explain the two main research questions of the study. In addition, we decided to remove the heading „Computer-Based Systems on Students‘ Cohesive Writing“ (see #12) from the manuscript since the concepts discussed in this section probably misled readers about the purpose of the manuscript.

#12: Computer-Based Systems on Students’ Cohesive Writing. I’m not sure what this section adds. Would it be sufficient to have a single paragraph providing a short overview of the idea of using a computer-based system for evaluating cohesion in the earlier section?

The section „Computer-Based Systems on Students’ Cohesive Writing“ introduced the reader to the concept of text cohesion and computer-based feedback systems. However, in line with comment #2 and #7, some information in this section was redundant with the introduction. Therefore, we removed the section „Computer-Based Systems on Students’ Cohesive Writing“ from the manuscript and incorporated the information on computer-based feedback in the introduction. Overall, we shortened the text segment on computer-based feedback systems to a single paragraph.

"Despite its central role for supporting readers’ comprehension, college students often face difficulties in writing cohesive texts [10,11]. Therefore, particularly students require ample formative feedback on the cohesion of their writing [12]. Providing instant feedback however is relatively time-consuming and often not feasible during regular teaching. Thus, recently, a variety of computer-based feedback systems have been developed to improve students’ writing for specific linguistic features such as text cohesion, particularly in the early stages of writing instruction [12–15]. The advantage of these systems is that they provide students with instant and time-independent information about the quality of their writing. Many of these systems generate graphical visualizations from students’ texts in the form of concept maps [16,17]. These concept maps provide students with an additional external representations of their text and direct their attention to distinct textual deficits in order to activate appropriate revision activities [15]."

#13: CohViz: A Feedback System to Support Students’ Cohesive Writing. I still would like more information about the setting for using this tool—who uses it?

As explained in comment #6, CohViz has so far mainly been used as a research tool in studies conducted by the authors. In these studies, we have used the tool both in controlled laboratory situations and in ecologically valid classroom contexts. However, since the utility of CohViz was not explained in sufficient detail in the manuscript, we have added a short section to the introduction in which we explain for which purposes CohViz is particularly valuable. 

"The key advantage of CohViz is that it supplements teachers’ feedback in that it allows to provide students with quick and specific feedback on the degree of cohesion of their texts. In practice, the feedback can be embedded as a homework assignment [21] or modelled by teachers to show textual deficits of non-cohesive texts [15]. For example, an instructor could use the tool to inform students why a particular text is high or low in cohesion (e.g., by showing texts which yield numerous fragments or texts which do not have central concepts with many relations). Similarly, students can use the tool to check short texts such as abstracts or summaries for cohesion during writing."

#14: I am unsure when and where the cited studies by Lachner and colleagues were completed. In a college? Recently? Why?

We added the information accordingly, see also comment #6. To date, five studies with CohViz have been published (in-between 2017-2019). The studies were conducted with German university students. The main reason for conducting these studies was students' difficulties in writing cohesive texts. Since writing cohesive texts is a central problem for students, we developed an automated feedback system to give them immediate feedback on the cohesion of their texts. Earlier work by Lachner and Nückles (2015) could show that the explanations of university students are more fragmented and thus less cohesive than the explanations of experts. In the revised manuscript, we added the information about the studies accordingly:

"So far, the effectiveness of CohViz has been tested in various settings between 2017 and 2019, including controlled laboratory studies and ecologically valid field studies in a wide range of disciplines (e.g. biology, ethics, philosophy, educational psychology, teacher education). In these experimental studies, the authors examined the effectiveness of CohViz with college students [15,20,21,29]. Overall, in a mini meta-analysis Burkhart, Lachner, and Nückles [30] could show that CohViz has a medium effect on both local and global cohesion and was thus effective in improving the cohesion of students’ texts (i.e., local cohesion g = 0.62; global cohesion g = 0.57). A potential explanation for the obtained effects of CohViz can be found in the think-aloud study by Lachner, Burkhart, Nückles [15]. The think-aloud study could show that students who processed the concept maps could infer both local and global writing plans from the concept maps directly. In addition, the analysis of the concept map triggered negative monitoring processes (i.e., student thought about the macrostructure of a text) which lead to further planning processes."

Additionally, due to the comment, we became aware that some of the references were not updated correctly due to technical issues. For example, in the manuscript we incorrectly credited Temperley as a reference for CohViz. We carefully revised the manuscript accordingly. Thank you again, for making us aware for potential flaws in the references. 

#15: Accuracy as a Critical Component of Effective Computer-Based Feedback, Reliability, Validity. I’m not sure what these sections add. Ideally, the section about CohViz should simply state the limitations of the prior studies, the actual needs of users, and the need for the current studies much more concisely, following the style of the current overall discussion.

In the revised manuscript, we have followed the reviewer’s recommendations and removed the headings mentioned from the manuscript. In the introduction, we first provide a thorough introduction to CohViz, its efficacy and applicability, and then present previous research on its effectiveness. We then go straight to the limitations of previous studies and highlight that the basic assumptions of reliability and validity have not been rigorously tested so far. 

#16: Overview of the Present Studies. I would not describe the studies as “empirical”. 

In the revised manuscript we refrained from using the term „empirical“ and now speak of a „corpus-based study“. 

#17: I would describe the studies in terms of what they are testing (“reliability” and “validity”) instead of calling them Study 1 and Study 2 and expecting the reader to keep track of the shifts.

In the revised manuscript we changed to labelling of the studies to Study 1 -> reliability study and Study 2 -> validity study.

#18: Study 1 Methods: I find this section confusing, largely because I’m not sure who is doing what where and when and to which corpus of texts. Ideally the methods would identify in order: The goal of the study (assess reliability of concept maps between machine and human coders); How human coders were identified; Which corpus of work was used for CohViz versus human coders. How was the full corpus generated and in which setting and by whom? How was the smaller representative sample generated in which setting and why whom?; How were the outputs generated? Who reviewed them?

Indeed, the description of the process was not clearly laid out in the manuscript. Therefore, with regard to the reviewer’s comment, we have tried to address each question raised by the reviewer in the corpus section of the first study on reliability (see p. 8). In addition, to increase the comprehensibility of the methods section, we have added a figure (Fig 2) in which we depict the data analysis procedure for each study individually. With regard to the ordering of the methods section, as mentioned in comment #3, we have followed the style of previous studies published in PlosOne and report the subsections in the following order: “Corpora collection”, “Measures”, and “Data Analysis”.

#19: I would move the explanation about how CohViz works to the introduction.

We followed the recommendations by the reviewer and moved to section on how CohViz works to the introduction (see heading “Generation of the CohViz concept maps”). 

#20: I would strongly recommend avoiding the use of “Lincoln freed the slaves” as an example. It would be highly offensive to many US readers. Please choose something more neutral. Maybe “Hollywood baked the rolls”?

We changed to example and now report the example provided by the reviewer.

#21: Study 1: Results and discussion: Please move the statistical methods back to the methods section

In the inspection of the manuscript we saw that the power analyses for both studies were presented in the results section. Accordingly, we have moved these power analyses to the analysis sections of the method.

#22: This discussion is far too long and has far too many little asides and explanations. As a reader, it is helpful to have the results presented as: How big were the samples reviewed? Did the representative sample for human coders adequately mimic the overall sample? How did you know that? What actually was completed? How many concept maps were generated and reviewed? How well did the two methods correlate?

We followed suggestions by comment #3 and now provide only a general discussion of both studies. Therefore, we removed the discussion of Study 1 (reliability study) from the manuscript.

#23: Study 2: This seems very long and not clear because information that should be in the introduction is interspersed throughout the text, there are methods in the results section and too much explanation as the results are presented. It simply feels very similar to the problems I noted in Study 1, where the methods and results are not clearly laid out in a logical order, but rather seem to demand that the reader carefully track with the researchers.

As mentioned in #18, we have thoroughly restructured the methods and results section of the manuscript. To increase the comprehensibility of the results, we divided this section into a subsection on reliability and validity (see also comment #3).

#24: Overall discussion: This is generally well written and concise. Please describe the studies in terms of what they tested and not by numbers. It makes the work very hard to read.

As mentioned in #17, we have renamed the studies to Study 1 -> reliability study and Study 2 -> validity study.

#25: Limitations and future directions: This seems very wordy and a bit too speculative. I applaud the correctness of the limitations identified, but the lengthy caveats and explanations are not really helpful. More helpful are the practical modes of moving forward.

In the revised manuscript, we have shortened the technical limitations of the current study and added a practical limitation which, to our knowledge, would be a logical next step to improve the utility of the feedback for improving students’ ability to write cohesive texts.

 

Reviewer #2: 

#1: The authors showed that CohViz is a liable and valid approach to provide students with feedback on the cohesion of their writing. A major cause which might lead to an almost-reject decision is that their research purpose is not convincing. Specifically, the authors claim at the beginning that the effectiveness of CohViz may come from a superficial warning or an essential improvement on cohesion. And they find at last that the effectiveness of CohViz comes from the essential improvement on cohesion. The reason behind their questioning that the effectiveness of CohViz may only come from a superficial warning is not solid. Generally speaking, CohViz-like systems are just developed from the idea that bad writing usually companies with bad cohesion. It is hard to believe that these systems do not benefit from the above design idea. As in the Aspirin example illustrated by authors themselves, the authors’ work in CohViz is like to proof that Aspirin is not a placebo. I believe this is not enough, in terms of innovation. If the authors can provide an example telling that some students do not actually improve cohesion suggested by CohViz-like systems, but make other changes which also improves the quality of writing, I will be more than happy to recommend their revisions to be accepted. Other problems are listed as follows.

We carefully thought about the comment. We agree that the tool is only capable to provide feedback on the coehison of texts. However, we want to note that besides other textual features such as lexical complexity, syntactic complexity, and text length, cohesion has been demonstrated to be a crucial textual feature which contributes to the comprehensibility of texts, and as such to writing quality (see MacArthur, Jennings, & Philippakos 2018; Wiley et la., 2017). In addition, previous research has shown that cohesion contributes to the overall comprehensibility of texts (McNamara, 2001; McNamara, Kintsch, Songer, & Kintsch, 1996; Ozur, Dempsey, & McNamara, 2007; O'Reilly & McNamara, 2007). Empirical research provided further evidence that students face difficulties in writing cohesive texts (Connor, 1984; Granger & Tyson, 1996; Hinkel, 2001; Zhang, 2000). The need to support students writing cohesive texts is also reflected by the fact that a plethora of writing guides at universities aim to improve students’ ability to write cohesive texts (e.g., Education Development Unit, 2002; Purdue Online Writing Lab, 2020; The University of Melbourne, 2020; Writing Center, 2020). Against this background, the central educational goal of CohViz is to improve students' ability to write cohesive texts. Previous research has confirmed the effectiveness of CohViz. For example, on page 6 of the revised manuscript we report the results of a mini meta-analysis which has shown that CohViz has a medium effect on the local and global cohesion of students texts. Unfortunately, with regard to comment # 5, some references, including references to the effectiveness of CohViz, were incorrect because the reference list was not updated correctly. We have corrected this error in the revised manuscript and now report on previous research on CohViz correctly. 

The central purpose of the study results from the fact that previous studies have not addressed the central question of the reliability and validity of feedback. Although we know the (meta-)cognitive processes triggered by the feedback, we have missed the fundamental basis for these processes, namely whether the feedback provides students with accurate feedback on the textual cohesion. These results may help to better understand the chain of effects of the feedback. In the field of automated writing evaluation systems, such studies are of utmost importance in order to establish the credibility of the system (Allen, 2016; Attali, 2013). What automated writing evaluation systems and computer-based feedback systems have in common is that they have to diagnose certain textual deficits from the texts. The exact representation of these deficits is therefore of central importance for the feedback system. In fact, most computer-based feedback systems rely directly on automated text evaluation systems that have been tested for reliability and validity (see Roscoe & McNamara, 2013; Wade-Stein & Kintsch, 2004) to provide students with feedback on the quality of their texts. The current study therefore closes this research gap and better helps to understand the chain of effects of concept map feedback to improve students’ ability to write cohesive texts. 

#2: Second Paragraph in Introduction: The first question is: Which references among [4,5,10,11] do the authors believe to document CohViz is a beneficial tool? I believe there is none, otherwise they ought to provide a specific reference indicating where Figure 1 comes from. I suggest that the authors first define CohViz as a family/cluster of computer-based feedback systems which help students’ writing using cohesion. If CohViz is really a specific tool, then the success of W-Pal (the real system developed in [10]) does not necessarily mean CohViz is also a success. At least, the authors may add that CohViz is developed based on a duplication of W-Pal.

Indeed, some references in the manuscript were not correct as the reference list was not updated correctly. For example, in the issue raised by the reviewer, we mistakenly credited Temperley as a reference for CohViz. As mentioned in #1, we have already conducted several studies on the effectiveness of CohViz. In a comprehensive meta-analysis, for example, we were able to show that CohViz has a medium effect on both the local and global cohesion of students‘ texts (Burkhart, Lachner, & Nückles, 2020). We have carefully corrected these references in the revised manuscript and hope that all references are given correctly. 

#3: The second question is: I understand the purpose of authors using the Aspirin example, but it is far from CohViz. The authors may simply indicate that they know CohViz is effective, but do not know how it works or do not know the detail mechanism makes CohViz effective. That is enough. NO need to use an irrelevant example to point out the logic error (even do not need to mention the logic error because it may probably be the authors’ own logic, not others’). Other readers may simply want to know more, e.g., why is it effective? If this is true, then there is actually no logic error. Please go quick and straight to the point. 

In the revised manuscript, we have removed the information on aspirin from the manuscript and tried to make it clear that the main research question of the current study was to investigate how reliable and valid the feedback is in terms of text cohesion. We have also removed the sentence about the logic error from the section.

#4: And I am also disappointed when reading the conclusion of this manuscript that it only confirms CohViz’s effectiveness, but seemingly forgets the why question. To sum up, the 2nd paragraph in Introduction needs to be re-written.

We agree with the reviewer that the why question has not been sufficiently addressed in the manuscript. We therefore added a section on the (meta-)cognitive processes triggered by the feedback to the introduction (see p. 6).

#4: 2) Page 5, the first sentence under the subtitle “CohViz: a feedback system to support students’ cohesive writing”. I did not find any “CohViz” in [5]. Do you mean [5] developed “CohViz”? I believe not.

This comment directly relates to #2. We corrected the mistake in the revised manuscript.

#5: 3) Page 6, Lachner, Burkhart, and Nückles are not the authors of [5], and they are also not the authors of [4].

This comment directly relates to #2. We corrected the mistake in the revised manuscript.

#6: 4) Related work is not reviewed adequately. Other works can be used as a start point, such as [5]. The authors may tell readers what [5] has already achieved in the mechanism investigation.

This comment directly relates to #2. We corrected the mistake in the revised manuscript.

#7: 5) The common senses of reliability and validity do not need to be re-introduced.

6) 

We have followed the recommendations of the reviewer and removed the general description of reliability and validity from the manuscript. 

#8: The demographic information of students and human experts is missing.

In the methods section of the revised manuscript, we have added the mean age of the human expert raters and the standard deviation. Regarding the demographic information of the students from the corpus used for the reliability study, we now report that they were college students from German universities. However, since the sample was randomly drawn from 901 texts, we are not able to report the exact mean age of the students (see p. 13):

"To compare the CohViz concept maps to concept maps generated from human expert raters, we asked four human expert raters to segment the texts from the reliability corpus into propositions as a basis for the generation of the concept maps (see Fig 4 for the full processing of the corpus). Therefore, we aimed for experienced raters with a solid background in linguistics who had previous experience in propositional segmentation. Among the raters who fulfilled these criteria, we asked four advanced master students with a major in applied linguistics or learning and instruction to analyze the corpus. All raters came from the same German university as the authors. Their mean age was 24 (SD = 2.94). They were already familiar with the procedure of propositional segmentations since propositional segmentation was part of their studies’ curriculum. To ensure a uniform prior knowledge on the procedure, each rater was provided with multiple in-depth training sessions (five hours on average) on propositional segmentation and text cohesion. In these training sessions, raters were instructed on different cohesion strategies (e.g., argument overlap, connectives, bridging information). Additionally, they were trained in propositional segmentation with authentic practice material."

 

Reviewer #3: 

#1: 1. Language problems: 1.1. hypernyms => hyperonyms (many times); 1.2. funders => founders

We carefully revised the manuscript for language problems and had an English native speaker proofread the entire manuscript. In the literature there is a distinction between the terms hypernym and hyponym. Hyponyms refer to subordinate concepts (e.g., red in relation to color) whereas hypernyms refer to superordinate concepts (e.g., color in relation to red). We introduced the term hypernym in the section on „Word concreteness“. In this section, we argue that an abstract word is defined by the fact that it has more hypernyms than a non-abstract word. We think, therefore, that the use of the concept hypernym in this case is warranted. The concept of hyponyms has only been mentioned once as an example of a cohesion device between adjacent sentences (see „Generation of the human expert concept maps“). However, to make it more comprehensible, we now speak of subordinate concepts. As for the terms funders and founders we have not found any occurences in the manuscript. Nevertheless, we checked the entire manuscript for potential flaws and corrected them if necessary.

#2: 2. Presentation problems: 2.1 Table 1: the table title nor its header do not specify the meaning of values in brackets

In fact, the explanation of the values in brackets was missing in Table 1. We have therefore added a note explaining that these values refer to standard deviations: „aValues in brackets refer to standard deviations.“

#3: 2.2. The format of references is not uniform (others => et al.)

We went through the manuscript carefully and used et al. for all references where appropriate for the Vancouver style.

#4: 2.3 Fig 1., Fig 3., the quality is low, please consider vector format

We have redesigned all figures to ensure the quality of the figures.

#5: 3. Other problems: 3.1 some of the the analyzed measures were not defined formally, e.g. "mean readability score based on the Flesch-Kincaid measure",

"Adjacent semantic overlap", "Word concreteness"; e.g. "Adjacent semantic overlap measures the cosine similarity between neighboring sentences, and text level semantic overlap measures the cosine similarity between all possible sentence pairs of the text." is not explaining how the measure is calculated.

Indeed, the Flesch-Kincaid measure was not introduced in the manuscript. Hence, we added a short explanation what it measures and how to interpret the measure (see p. 9). As for the divergent and convergent measures (e.g., adjacent semantic overlap), we thoroughly explained how these measures were generated in the section on „Measures of convergent and divergent validity“. In addition, we added an explanation for the measures root-type-token ratio and semantic overlap in Fig 4. However, if the reviewer insists to see are more detailed explanation of how these measures were generated, we will be happy to provide them.

#6: 3.2 Obtained results can be biased by the selection of the human experts. The process of their selection is not given.

Even though there is evidence that raters differ in their rating behaviors (Eckes, 2008), the results from our interrater reliability indicate that the raters showed a high degree of consistency in segmenting the corpus into propositions (ICC > .76). However, to avoid potential biases, we also added additional safeguards. First, we aimed to find expert raters with a solid background in linguistics who had prior experience with propositional segmentation. Addtionally, these raters received in-depth trainings on how to segment texts into propositions (see text segment below). In addition, we chose to recruit four expert raters in total. In other studies using expert raters, it is common practice to use only two raters (see Callender & McDaniel, 2009; Chi et al., 2018; Müller & Oppenheimer, 2014; Ozuro, Dempsey, & McNamara, 2009). We therefore believe that a total of four raters is above the current standard and given our selection process we are confidend that the results are not biased. Nevertheless, we added crucial information about the selection process of the human expert raters to the manuscript.

#7: 3.3. It would be good to give information who is the author of CohViz system. In the actual form of the text it is unclear.

Unfortunately, some references in the manuscript were wrong as the reference list was not updated correctly. For example, in the issue raised by the reviewer, we mistakenly credited Temperley as a reference for CohViz. Hence, it was not clear who the auhors of CohViz are. We have carefully corrected these references in the revised manuscript and hope that the authors of CohViz can now be correctly identified (e.g., Lachner, Burkhart, & Nückles, 2017).

#8: Summary: The language is good. The quality of illustrations is low and the presentation should be improved. But what is most important area that needs fixing includes definitions of analyzed measures and analysis of the characteristics of used population of human experts.

Recommendation: major rework

 

References

Abu-Shaheen, A., Yousef, S., Riaz, M., Nofal, A., AlFayyad, I., Khan, S., & Heena, H. (2018). Testing the validity and reliability of the Arabic version of the painDETECT questionnaire in the assessment of neuropathic pain. PLoS ONE, 13(4), 1–13. https://doi.org/10.1371/journal.pone.0194358

Allen, L. K., Jacovina, M. E., & McNamara, D. S. (2016). Computer-Based Writing Instruction. In C. A. MacArthur, S. Graham, & J. Fitzgerald (Eds.), Handbook of Writing Research (2nd ed., pp. 316–329). New York, London: The Guilford Press.

Attali, Y. (2013). Validity and Reliability of Automated Essay Scoring. In M. D. Shermis & J. Burstein (Eds.), Handbook of Automated Essay Evaluation: Current Applications and New Directions (pp. 181–198). Routledge.

Callender, A. A., & McDaniel, M. A. (2009). The limited benefits of rereading educational texts. Contemporary Educational Psychology, 34(1), 30–41. https://doi.org/10.1016/j.cedpsych.2008.07.001

Center, W. (2020). Flow and Cohesion. Retrieved from https://www.umass.edu/writingcenter/flow-and-cohesion

Chi, M. T. H., Adams, J., Bogusch, E. B., Bruchok, C., Kang, S., Lancaster, M., … Yaghmourian, D. L. (2018). Translating the ICAP Theory of Cognitive Engagement Into Practice. Cognitive Science, 42(6), 1777–1832. https://doi.org/10.1111/cogs.12626

Connor, U. (1984). A study of cohesion and coherence in English as a second language students’ writing. Paper in Linguistics, 17(3), 301–316. https://doi.org/10.1080/08351818409389208

Granger, S., & Tyson, S. (1996). Connector usage in the English essay writing of native and non-native EFL speakers of English. World Englishes, 15(1), 17–27. https://doi.org/ 10.1111/j.1467-971X.1996.tb00089.x

Hara, N., Matsudaira, K., Masuda, K., Tohnosu, J., Takeshita, K., Kobayashi, A., … Kikuchi, N. (2016). Psychometric assessment of the Japanese version of the zurich claudication questionnaire (ZCQ): Reliability and validity. PLoS ONE, 11(7), 1–10. https://doi.org/ 10.1371/journal.pone.0160183

Hinkel, E. (2001). Matters of cohesion in L2 academic texts. Applied Language Learning, 12(2), 111–132.

Kogure, T., Sumitani, M., Suka, M., Ishikawa, H., Odajima, T., Igarashi, A., … Kawahara, K. (2014). Validity and reliability of the Japanese version of the newest vital sign: A preliminary study. PLoS ONE, 9(4), 1–6. https://doi.org/10.1371/journal.pone.0094582

Kruizinga, I., Jansen, W., de Haan, C. L., van der Ende, J., Carter, A. S., & Raat, H. (2012). Reliability and validity of the dutch version of the brief infant-toddler social and emotional assessment (BITSEA). PLoS ONE, 7(6). https://doi.org/10.1371/journal.pone.0038762

Lab, P. O. W. (2020). Revising for Cohesion. Retrieved from https://owl.purdue.edu/owl/general_writing/the_writing_process/proofreading/revising_for_cohesion.html

Meisuo, Z. (2000). Cohesive Features in the expository writing of undergraduates in two chinese universities. RELC Journal, 31(1), 61–95.

Melbourne, U. of. (2020). Improving cohesion. Retrieved from https://services.unimelb.edu.au/__data/assets/pdf_file/0011/1264790/Improving_cohesion_Update_051112.pdf

Mueller, P. A., & Oppenheimer, D. M. (2014). The Pen Is Mightier Than the Keyboard. Psychological Science, 25(6), 1159–1168. doi:10.1177/0956797614524581

Ozuru, Y., Dempsey, K., & McNamara, D. S. (2009). Prior knowledge, reading skill, and text cohesion in the comprehension of science texts. Learning and Instruction, 19(3), 228–242. https://doi.org/10.1016/j.learninstruc.2008.04.003

Unit, E. D. (2020). Editing your Writing for Content, Coherence and Cohesion. Retrieved from http://wwwdocs.fce.unsw.edu.au/fce/EDU/educoncohcoh.pdf

Roscoe, R. D., & McNamara, D. S. (2013). Writing pal: Feasibility of an intelligent writing strategy tutor in the high school classroom. Journal of Educational Psychology, 105(4), 1010–1025. https://doi.org/10.1037/a0032340

Steenson, S., Özcebe, H., Arslan, U., Ünlü, H. K., Araz, Ö. M., Yardim, M., … Huang, T. T. K. (2018). Assessing the validity and reliability of family factors on physical activity: A case study in Turkey. PLoS ONE, 13(6), 1–15. https://doi.org/10.1371/journal.pone.0197920

Wade-Stein, A. D., & Kintsch, E. (2004). Summary Street: Interactive Computer Support for Writing. Cognition and Instruction, 22(3), 333–362.

Wiley, J., Hastings, P., Blaum, D., Jaeger, A. J., Hughes, S., Wallace, P., … Britt, M. A. (2017). Different Approaches to Assessing the Quality of Explanations Following a Multiple-Document Inquiry Activity in Science. International Journal of Artificial Intelligence in Education, 27(4), 758–790. https://doi.org/10.1007/s40593-017-0138-z

---

## [Decision Letter · Decision Letter 1]

22 May 2020

PONE-D-19-33124R1

Assisting students’ writing with computer-based concept map feedback: A validation study of the CohViz feedback system

PLOS ONE

Dear Dr. Burkhart,

Thank you for submitting your manuscript to PLOS ONE. It was reviewed by the three Reviewers including me as an Academic Editor (reviewer #3). After careful consideration, we feel that it has merit but does not fully meet PLOS ONE’s publication criteria as it currently stands. Therefore, we invite you to submit a revised version of the manuscript that addresses the points raised during the review process.

In particular:

PLOS One data and software availability criteria should be met,quantitative comparison of selected measures of CohViz and other tools for analyses of writing quality considered by Authors should be clearly presented,formal definitions of methods and analyzed measures would help to reproduce the results.

We look forward to receiving your revised manuscript.

Kind regards,

Maciej Huk, Ph.D.

Academic Editor

PLOS ONE

Reviewers' comments:

Reviewer's Responses to Questions

**Comments to the Author**

1. If the authors have adequately addressed your comments raised in a previous round of review and you feel that this manuscript is now acceptable for publication, you may indicate that here to bypass the “Comments to the Author” section, enter your conflict of interest statement in the “Confidential to Editor” section, and submit your "Accept" recommendation.

Reviewer #1: (No Response)

Reviewer #2: All comments have been addressed

Reviewer #3: (No Response)

2. Is the manuscript technically sound, and do the data support the conclusions?

Reviewer #1: Yes

Reviewer #2: Yes

Reviewer #3: Partly

3. Has the statistical analysis been performed appropriately and rigorously? 

Reviewer #1: Yes

Reviewer #2: Yes

Reviewer #3: I Don't Know

4. Have the authors made all data underlying the findings in their manuscript fully available?

Reviewer #1: Yes

Reviewer #2: Yes

Reviewer #3: No

5. Is the manuscript presented in an intelligible fashion and written in standard English?

Reviewer #1: Yes

Reviewer #2: Yes

Reviewer #3: Yes

6. Review Comments to the Author

Reviewer #1: The paper as initially submitted was already a valuable addition to the literature and based on a well-designed study. Therefore the prior comments largely had to do with how the information was presented as well as the completeness of small details that are necessary to understand how the study was designed. This is an excellent writing revision, following my recommendations very well while also addressing the concerns of reviewer two.

More importantly, this rewrite made it possible to see that the statistics are well-done. It is evident that the rigor is adequate to answer the researchers' questions. This paper now provides more adequate information to help readers and researchers consider how they might use this technology in various academic settings.

Reviewer #2: (No Response)

Reviewer #3: >>> 1. Language problems: not detected

>>> 2. Presentation problems:

2.1 The functioning of the CohViz system is mainly textual. It would be good to present mathematical formulas used to calculate important measures of the text cohesion features used by the CohViz.

>>> 3. Other problems:

3.1 The link to data given by the Authors is invalid.

Authors write: "All files are available on github at the following address:

" ext-link-type="uri" xlink:type="simple">https://github.com/ch-bu/dataassisting-students-writing-with-computer-based-concept-map-feedback"

This link goes to "Error 404 - page not found"

3.2 It would be good to give direct information who is the author of CohViz system. In the actual form of the text it is still unclear without reading the text of given references (e.g. [15]). Please consider the following change:

CohViz is a graphical feedback which automatically provides students with concept maps as external representations to improve the cohesion of their writing (see Fig 1) [15,20,21].

=

CohViz is a graphical feedback tool we developed, which automatically provides students with concept maps as external representations to improve the cohesion of their writing (see Fig 1) [15,20,21].

3.3 The main focus of the manuscript is the CohViz system developed by the Authors. Manuscript includes description of the system and validation of its reliability and accuracy. In such case PLOS One Software availability criteria should be met:

"PLOS ONE will consider submissions that present new methods, software, databases, or tools as the primary focus of the manuscript if they meet the following criteria:

(...)

Validation: Submissions presenting methods, software, databases, or tools must demonstrate that the new tool achieves its intended purpose. (...)

Availability: If the manuscript’s primary purpose is the description of new software or a new software package, this software must be open source, deposited in an appropriate archive, and conform to the Open Source Definition. (...)

Please see:

https://journals.plos.org/plosone/s/submission-guidelines#loc-methods-software-databases-and-tools

Validation is presented. Availability is not met.

3.4 Authors write:

"In the validity study, we demonstrated the convergent and divergent validity of the CohViz system in comparison to measures of other well-established research tools for writing quality [35,43]"

Can Authors point out where in their manuscript the "measures of other well-established research tools for writing quality" are presented and compared with results for CohViz? Any comparison of values of selected measures for CohViz and considered "other" tools? Any table or figure?

 Summary: The language is good. The presentation should be improved. Data availability and software availability are not met. But what is most important area that needs fixing includes formal definitions of analyzed measures and comparison with other similar tools.

Recommendation: major rework

7. PLOS authors have the option to publish the peer review history of their article (what does this mean?). If published, this will include your full peer review and any attached files.

Reviewer #1: Yes: Lisa DeTora

Reviewer #2: Yes: Tai Wang

Reviewer #3: No

---

## [Author Response · Author response to Decision Letter 1]

4 Jun 2020

Reviewer #1: 

The paper as initially submitted was already a valuable addition to the literature and based on a well-designed study. Therefore the prior comments largely had to do with how the information was presented as well as the completeness of small details that are necessary to understand how the study was designed. This is an excellent writing revision, following my recommendations very well while also addressing the concerns of reviewer two.

More importantly, this rewrite made it possible to see that the statistics are well-done. It is evident that the rigor is adequate to answer the researchers' questions. This paper now provides more adequate information to help readers and researchers consider how they might use this technology in various academic settings.

Thank you very much. We appreciate your feedback.

Reviewer #2: 

(No Response)

Reviewer #3: 

1. Language problems: not detected

2. Presentation problems:

2.1 The functioning of the CohViz system is mainly textual. It would be good to present mathematical formulas used to calculate important measures of the text cohesion features used by the CohViz.

Thank you for the suggestion. Under the sub-chapter Structural indicators of the concept maps we have now added a formal definition of the three central structural indicators of CohViz using graph theory notation (p. 15). We now describe the fragments, relations, and concepts in the language of graph theory and explain these definitions using the concept map in Figure 3 as an example. At the end of the sub-chapter we also refer to the public repository under which the algorithms (written in Python) for calculating these indicators can be found. Besides, we have added and described formulas that explain the calculation of the measures of adjacent argument overlap and text level overlap (p. 16-17).

3. Other problems:

3.1 The link to data given by the Authors is invalid. Authors write: "All files are available on github at the following address: https://github.com/ch-bu/dataassisting-students-writing-with-computer-based-concept-map-feedback". This link goes to "Error 404 - page not found"

We apologize for any inconvenience caused by the incorrect link. We have now uploaded the data and the analyses of the data to a public repository under the following address: https://doi.org/10.17605/OSF.IO/UHPF3 on the OSF portal. We have also published the algorithms we used to calculate the structural indicators of CohViz. The files for these calculations can be found in the analysis/stuctural_indicators directory of the repository. Also, a reference to the repository can be found in the manuscript on page 15.

3.2 It would be good to give direct information who is the author of CohViz system. In the actual form of the text it is still unclear without reading the text of given references (e.g. [15]). Please consider the following change: CohViz is a graphical feedback which automatically provides students with concept maps as external representations to improve the cohesion of their writing (see Fig 1) [15,20,21]. => CohViz is a graphical feedback tool we developed, which automatically provides students with concept maps as external representations to improve the cohesion of their writing (see Fig 1) [15,20,21].

We have incorporated the reviewer's suggestion and now report that we are the authors of the system (see the revised manuscript, p. 4).

3.3 The main focus of the manuscript is the CohViz system developed by the Authors. Manuscript includes description of the system and validation of its reliability and accuracy. In such case PLOS One Software availability criteria should be met:

"PLOS ONE will consider submissions that present new methods, software, databases, or tools as the primary focus of the manuscript if they meet the following criteria:

(...) Validation: Submissions presenting methods, software, databases, or tools must demonstrate that the new tool achieves its intended purpose. (...) Availability: If the manuscript’s primary purpose is the description of new software or a new software package, this software must be open source, deposited in an appropriate archive, and conform to the Open Source Definition. (...) Please see:

https://journals.plos.org/plosone/s/submission-guidelines#loc-methods-software-databases-and-tools. Validation is presented. Availability is not met.

In fact, we did not include a reference to the CohViz system in the manuscript. Accordingly, on page 4 of the revised manuscript, we have now included the reference to the system, which has been uploaded to the OSF public repository under the following address: https://doi.org/10.17605/OSF.IO/SA53E. We chose to publish the system under the MIT license as it conforms to the Open Source Definition (https://opensource.org/licenses) recommended by PLOS One. Under the MIT license, the software is freely available and may be freely distributed and modified.

3.4 Authors write:

"In the validity study, we demonstrated the convergent and divergent validity of the CohViz system in comparison to measures of other well-established research tools for writing quality [35,43]"

Can Authors point out where in their manuscript the "measures of other well-established research tools for writing quality" are presented and compared with results for CohViz? Any comparison of values of selected measures for CohViz and considered "other" tools? Any table or figure?

Indeed, the sentence was written misleadingly. By "other" we referred to the measures we reported in the chapter on convergent and divergent measures of text cohesion, which are reported in Table 2. We have revised the sentence, removed the word “other” and now report that we have used convergent and divergent measures from well-established assessment systems (see p. 27). Besides, we have added a box to Figure 2, in which we explain the processing of the Wikipedia corpus, and explicitly refer to the assessment system used by Hancke et al. During the revision we also noticed that we have used two terms to describe these systems in the manuscript: computer-linguistic frameworks and assessment systems. We now speak uniformly of assessment systems.

Summary: The language is good. The presentation should be improved. Data availability and software availability are not met. But what is most important area that needs fixing includes formal definitions of analyzed measures and comparison with other similar tools.

Recommendation: major rework

---

## [Decision Letter · Decision Letter 2]

11 Jun 2020

Assisting students’ writing with computer-based concept map feedback: A validation study of the CohViz feedback system

PONE-D-19-33124R2

Dear Dr. Burkhart,

We’re pleased to inform you that your manuscript has been judged scientifically suitable for publication and will be formally accepted for publication once it meets all outstanding technical requirements. The acceptance was suggested by the three Reviewers: two "accept" decisions for version R1 and third "accept" by me as an Academic Editor for version R2.

Kind regards,

Maciej Huk, Ph.D.

Academic Editor

PLOS ONE

Additional Editor Comments (optional):

Reviewers' comments:

Reviewer's Responses to Questions

**Comments to the Author**

1. If the authors have adequately addressed your comments raised in a previous round of review and you feel that this manuscript is now acceptable for publication, you may indicate that here to bypass the “Comments to the Author” section, enter your conflict of interest statement in the “Confidential to Editor” section, and submit your "Accept" recommendation.

Reviewer #3: All comments have been addressed

2. Is the manuscript technically sound, and do the data support the conclusions?

Reviewer #3: Yes

3. Has the statistical analysis been performed appropriately and rigorously? 

Reviewer #3: I Don't Know

4. Have the authors made all data underlying the findings in their manuscript fully available?

Reviewer #3: (No Response)

5. Is the manuscript presented in an intelligible fashion and written in standard English?

Reviewer #3: Yes

6. Review Comments to the Author

Reviewer #3: 1. Language problems:

1.1 "for the full computer algorithm" = "for the full algorithm" / "for the full source code of the CohViz program"

 2. Presentation problems: not detected

 3. Other problems:

3.1 It would be good to extend the text with comparison of the CohViz with other systems supporting cohesive writing.

 Summary:

Exept minor problems the language is good and thoughts are presented clearly. Presentation of the CohViz system is detailed and the software is available freely as OpenSource.

Comparison of described system with other similar tools would increase the value of the text.

Recommendation: accept

===EOT===

7. PLOS authors have the option to publish the peer review history of their article (what does this mean?). If published, this will include your full peer review and any attached files.

Reviewer #3: No

---

## [Editor Report · Acceptance letter]

17 Jun 2020

PONE-D-19-33124R2 

Assisting students’ writing with computer-based concept map feedback: A validation study of the CohViz feedback system 

Dear Dr. Burkhart:

I'm pleased to inform you that your manuscript has been deemed suitable for publication in PLOS ONE. Congratulations! Your manuscript is now with our production department. 

Kind regards, 

on behalf of

Dr. Maciej Huk 

Academic Editor

PLOS ONE